# Double Randomized Underdamped Langevin with Dimension-Independent Convergence Guarantee

**Yuanshi Liu**[*], **Cong Fang**[✉*], **Tong Zhang**[†]

[*] School of Intelligence Science and Technology, Peking University
[†] Department of Computer Science & Engineering, the Hong Kong University of Science and Technology
{liu_yuanshi, fangcong}@pku.edu.cn, tongzhang@tongzhang-ml.org

## Abstract

This paper focuses on the high-dimensional sampling of log-concave distributions with composite structures: $p^*(\mathrm{d}\mathbf{x}) \propto \exp(-g(\mathbf{x}) - f(\mathbf{x}))\mathrm{d}\mathbf{x}$. We develop a double randomization technique, which leads to a fast underdamped Langevin algorithm with a dimension-independent convergence guarantee. We prove that the algorithm enjoys an overall $\widetilde{\mathcal{O}}\left(\frac{(\mathrm{tr}(H))^{1/3}}{\epsilon^{2/3}}\right)$ iteration complexity to reach an $\epsilon$-tolerated sample whose distribution $p$ admits $W_2(p, p^*) \leq \epsilon$. Here, $H$ is an upper bound of the Hessian matrices for $f$ and does not explicitly depend on dimension $d$. For the posterior sampling over linear models with normalized data, we show a clear superiority of convergence rate which is dimension-free and outperforms the previous best-known results by a $d^{1/3}$ factor. The analysis to achieve a faster convergence rate brings new insights into high-dimensional sampling.

## 1 Introduction

Sampling from a high-dimensional distribution serves as one of the key components in statistics, machine learning, and scientific computing, and constitutes the foundation of the fields including Bayesian statistics and generative models [Liu and Liu, 2001, Brooks et al., 2011, Song et al., 2020]. Recently, there is an emerging trend in designing provably faster Markov Chain Monte Carlo (MCMC) algorithms using techniques from first-order optimization [Dalalyan, 2017, Durmus et al., 2019, Cheng and Bartlett, 2018, Vempala and Wibisono, 2019, Chewi et al., 2021]. One typical MCMC algorithm that allows following the idea is the Langevin-type algorithms, which are of the central interest of this paper.

Langevin-type algorithms originated in statistical physics discretize a stochastic differential equation with stationary distribution corresponding to the target. For Gibbs distribution $p(\mathbf{x}) \propto e^{-U(\mathbf{x})}$, the standard overdamped Langevin algorithm uses Euler Maruyama scheme to discretize the diffusion process $\mathrm{d}\mathbf{x}_t = -\nabla U(\mathbf{x}_t)\mathrm{d}t + \sqrt{2}\mathrm{d}\mathbf{B}_t$. The algorithm iteratively performs the updates as for $n = 0, 1, 2, \ldots,$

$$\mathbf{x}_{n+1} = \mathbf{x}_n - h\nabla U(\mathbf{x}_n) + \sqrt{2h}\boldsymbol{\epsilon}_n, \tag{1.1}$$

where $\boldsymbol{\epsilon}_n \sim \mathcal{N}(0, I)$ follows the normal distribution and $h > 0$ is the step size.

This paper focuses on the dimension dependence of the convergence behavior of Langevin-type algorithms. Specifically, we consider sampling problems over potentially high-dimensional and strongly log-concave distributions with a composite structure: $p \propto e^{-U(\mathbf{x})} = e^{-g(\mathbf{x})-f(\mathbf{x})}$ where $g(\mathbf{x}) = \frac{m}{2}\|\mathbf{x}\|^2$. Regarding $p$ as a posterior distribution, $e^{-g(\mathbf{x})}$ corresponds to a Gaussian prior and $e^{-f(\mathbf{x})}$ corresponds to the likelihood. This structure also includes general $m$-strongly convex

potential functions $U(\mathbf{x})$, which can be split into a strongly convex term $g(\mathbf{x}) = \frac{m}{2}\|\mathbf{x}\|^2$ and a weakly convex one $f(\mathbf{x}) = U(\mathbf{x}) - \frac{m}{2}\|\mathbf{x}\|^2$.

The analysis of Langevin sampling from an optimization viewpoint may date back to Jordan et al. [1998]. The viewpoint has led to a surge of works that establish quantitative convergence guarantees for overdamped Langevin algorithms to sample log-concave distributions [Dalalyan, 2017, Durmus and Moulines, 2016]. It should be noted that these convergence guarantees always involve an extra $d$ dimension dependence due to the injection of non-negligible Gaussian noise, whereas the convergence rates of first-order optimization are often dimension-free [Nesterov, 2003]. Various accelerated methods have been proposed that can mitigate the dimension dependence and also achieve faster convergence. First, as the momentum acceleration of Langevin algorithms [Ma et al., 2021], underdamped Langevin Monte Carlo has a more stable trajectory and is known to exhibit a faster convergence rate [Cheng et al., 2018, Ma et al., 2021, Zhang et al., 2023]. Apart from using a different trajectory, more stable discretization schemes of the same diffusion process also lead to better convergence [Shen and Lee, 2019, Wibisono, 2019, He et al., 2020, Li et al., 2019]. However, the dimension dependence remains. To the best of our knowledge, the fastest randomized midpoint method for underdamped Langevin under Wasserstein distance achieves a $\mathcal{O}\left(\frac{d^{1/3}}{\epsilon^{2/3}}\right)$ convergence rate and still has a $d^{1/3}$ dimension dependence [Shen and Lee, 2019].

Recently another thread of works studies the convergence with weaker dimension dependence. Some researchers explore further general assumptions on the target distribution. The idea is by observing that some smoothness conditions can average out the dimension-dependent errors brought by the noise using Ito's formula. For example, Li et al. [2021] achieves a $\sqrt{d}$ dimension dependence with a 3-rd order growth condition. Another approach investigates the curvature of the target distribution and illustrates that dimension often does not determine the complexity of the sampling problem. Vono et al. [2022] propose an ADMM-type splitting algorithm with a dimension-free convergence rate when the likelihood is separable. Along the same thread, Freund et al. [2022] study the convergence rate of Langevin algorithms with no explicit dependence on dimension and propose to characterize the convergence rate by the upper bound of Hessian matrices of $f$. Specifically, by letting $H$ be the upper bound of the Hessian matrices of $f$, they show a variant of the overdamped Langevin algorithm achieves a convergence rate of $\mathcal{O}\left(\frac{\text{tr}(H)}{\epsilon}\right)$ in KL divergence. This result improves the rate under wide conditions because many high-dimensional sampling problems are intrinsically low-dimensional in the sense that $\text{tr}(H) = o(d)$, which frequently appears in machine learning. For example, when the potential function has a ridge separable structure with mild conditions, $\text{tr}(H)$ can be dimension-free (see Section 3.3 for more details).

Though these works succeed in obtaining a convergence rate of the Langevin algorithm with a weak dependence on the dimension, some important questions remain:

> *" How to design provably faster algorithms with weak dependencies on dimension even for the log-concave sampling?"*

Such a question is significant for understanding high-dimensional sampling and already includes lots of applications in real practice.

In this paper, we follow the regime of Freund et al. [2022] to design provably faster Langevin algorithms, which answers the above question affirmatively. We propose a double-randomized algorithm showing that variants of underdamped algorithms inherit the lower dimensional dependence in overdamped Langevin and a special design of random stepsize can still keep the property when using more stable discretization.

Specifically, we develop a double randomization technique and obtain the Double-Randomized Underdamped Langevin (DRUL) algorithm. We consider an averaged contraction effect to avoid dimension dependence. We design two distributions to achieve the mid-point acceleration with a randomized stepsize. DRUL is proven to enjoy an overall $\widetilde{\mathcal{O}}\left(\frac{(\text{tr}(H))^{1/3}}{\epsilon^{2/3}}\right)$ iteration and gradient complexity. For the posterior sampling over linear models, we show a clear superiority of DRUL, which achieves a dimension-free $\widetilde{\mathcal{O}}\left(\frac{1}{\epsilon^{2/3}}\right)$ convergence rate. The novel perspective of DRUL is introducing the random step size at each update. Such a random step size combined with a random midpoint point proposed by Shen and Lee [2019] reduces the discretization error and provides new insights into designing provably faster sampling algorithms.

Table 1: Comparison of the convergence rates for most related works. We consider the convergence in Wasserstein distance to a strongly log-concave and log-Lipschitz smooth target distribution (we summarize below the methods if additional assumptions are required). EU stands for Euler-Maruyama discretization and RMM stands for Randomized Midpoint discretization. 'Ridge-Separable Case' refers to the convergence rate when $f$ admits a ridge-separable structure (see (3.2)). Note that some results are established in the KL divergence. We convert these convergence rates into the ones in Wasserstein distance using Talagrand's inequality.

| Method | Convergence Rate | Ridge-Separable Case |
|---|---|---|
| Overdamped as Composite OptimizCRation [Durmus et al., 2019] (Proved in KL divergence) | $\mathcal{O}\left(\frac{d}{\epsilon^2}\right)$ | $\mathcal{O}\left(\frac{d}{\epsilon^2}\right)$ |
| Overdamped with EU [Li et al., 2021] (With an additional linear growth condition) | $\widetilde{\mathcal{O}}\left(\frac{\sqrt{d}}{\epsilon}\right)$ | $\widetilde{\mathcal{O}}\left(\frac{\sqrt{d}}{\epsilon}\right)$ |
| Underdamped with RMM [Shen and Lee, 2019] | $\widetilde{\mathcal{O}}\left(\frac{d^{1/3}}{\epsilon^{2/3}}\right)$ | $\widetilde{\mathcal{O}}\left(\frac{d^{1/3}}{\epsilon^{2/3}}\right)$ |
| Overdamped as Composite Optimization [Freund et al., 2022] (Proved in KL divergence) | $\mathcal{O}\left(\frac{\operatorname{tr}(H)+\|\mathbf{x}_*\|^2}{\epsilon^2}\right)$ | $\mathcal{O}\left(\frac{1}{\epsilon^2}\right)$ |
| **DRUL (Ours)** | $\widetilde{\mathcal{O}}\left(\frac{(\mathbf{tr}(\mathbf{H})+\|\mathbf{x}_*\|^2)^{1/3}}{\epsilon^{2/3}}\right)$ | $\widetilde{\mathcal{O}}\left(\frac{1}{\epsilon^{2/3}}\right)$ |

In summary, the contributions of the paper are listed below:

(A) We propose the Double-Randomized technique and design the DRUL algorithm.

(B) We show the DRUL converges to the target distribution in Wasserstein distance in $\widetilde{\mathcal{O}}\left(\frac{(\operatorname{tr}(H)+\|\mathbf{x}_*\|^2)^{1/3}}{\epsilon^{2/3}}\right)$ iterations. For posterior sampling over generalized linear models, a dimension-free $\widetilde{\mathcal{O}}\left(\frac{1}{\epsilon^{2/3}}\right)$ complexity can be achieved.

## 2 Related works

There has been a surge of works investigating the asymptotic guarantees for the Langevin-type algorithms [Roberts and Tweedie, 1996, Mattingly et al., 2002]. And a series of recent works establish the non-asymptotic quantitative analysis framework of the Langevin-type algorithms. For performance on strongly log-concave distributions, early works [Dalalyan, 2017, Durmus and Moulines, 2016] establish the non-asymptotic convergence rate in TV distance with Lipschitz gradients assumption for overdamped dynamics. Along the same setting, similar convergence rates are achieved in KL divergence using the Wasserstein gradient flow [Cheng and Bartlett, 2018, Durmus et al., 2019] or Wasserstein distance using the contractive property for overdamped Langevin. Underdamped Langevin accelerates the vanilla Langevin algorithms [Ma et al., 2021]. With the same setting mentioned above, a faster convergence rate can be established [Cheng et al., 2018, Dalalyan and Riou-Durand, 2020, Shen and Lee, 2019, Zhang et al., 2023] for underdamped Langevin algorithms. Beyond the log Lipschitz-smooth setting, other examples also show better results can be achieved, such as Durmus and Moulines [2019], Li et al. [2019] for overdamped Langevin algorithms. Previous works also investigate the convergence rate without strongly log-convex assumptions, such as the log-Sobolev inequality (LSI) condition which is similar to the Polyak-Łojasiewicz condition in optimization. With target distributions satisfying LSI and log-Lipschitz-smooth, the results can be extended for both overdamped Langevin [Vempala and Wibisono, 2019] and underdamped Langevin [Ma et al., 2021, Zhang et al., 2023].

Another thread of works explores the dimension dependence of Langevin-type algorithms. Despite the great similarity between the analysis in Langevin algorithms and optimization, the convergence of Langevin-type algorithms depends on dimension in the log-concave setting whereas convex optimization algorithms can often achieve dimension-independent results. With general strongly log-

concave (or LSI) and log-Lipschitz-smooth condition, previous works establish a $\mathcal{O}\left(\frac{d}{\epsilon^2}\right)$ convergence rate for overdamped algorithms [Durmus et al., 2019] and a $\widetilde{\mathcal{O}}\left(\frac{\sqrt{d}}{\epsilon}\right)$ convergence rate for underdamped algorithms [Cheng et al., 2018] to ensure finding a solution $x_n$ with $W_2(\text{Law}(\mathbf{x}_n), p) \leq \epsilon$ or $\sqrt{KL(\text{Law}(\mathbf{x}_n), p)} \leq \epsilon$. With an additional linear growth condition on third-order derivative, Li et al. [2021] achieve a $\widetilde{\mathcal{O}}\left(\frac{\sqrt{d}}{\epsilon}\right)$ convergence rate for overdamped Langevin. And for underdamped Langevin algorithms, Shen and Lee [2019] improve the dependence of dimension and obtains a $\widetilde{\mathcal{O}}\left(\frac{d^{1/3}}{\epsilon^{2/3}}\right)$ convergence rate. Recent work of Freund et al. [2022] characterizes the dimensional dependence of convergence rate by the upper bound of the Hessian matrix of $f$. As discussed in Section 1, Freund et al. [2022] establish a $\mathcal{O}\left(\frac{\text{tr}(H)+\|\mathbf{x}_*\|^2}{\epsilon}\right)$ convergence rate in KL divergence, which implies a $\mathcal{O}\left(\frac{\text{tr}(H)+\|\mathbf{x}_*\|^2}{\epsilon^2}\right)$ convergence rate in Wasserstein distance. And when $f$ admits a ridge-separable formula, Freund et al. [2022] obtain a dimension-free $\mathcal{O}\left(\frac{1}{\epsilon^2}\right)$ convergence rate with some mild assumptions. We summarize the comparison of these most related works in Table 2.

## 3  Preliminary and problem setup

### 3.1  Notations

We use the convention $\mathcal{O}\left(\cdot\right)$ and $\Omega\left(\cdot\right)$ to denote lower and upper bounds with a universal constant. $\widetilde{\mathcal{O}}(\cdot)$ ignores the polylogarithmic dependence. And use $f \lesssim g$ to denote $f = \mathcal{O}(g)$. Use $W_2(\mu, \nu)$ to denote the 2-Wasserstein distance of distribution $\mu$ and $\nu$. Use $\text{Law}(X)$ to denote the distribution of the random variable $X$. The Frobenius norm is denoted by $\|\cdot\|_F$ while $\|\cdot\|_2$ stands for operator 2-norm for matrices.

### 3.2  Sampling problem

We consider the distributions with a composite structure:

$$\mathrm{d}p(\mathbf{x}) \propto \exp\{-U(\mathbf{x})\}\mathrm{d}\mathbf{x} = \exp\left\{-g(\mathbf{x}) - f(\mathbf{x})\right\}\mathrm{d}\mathbf{x}, \tag{3.1}$$

where $g(\mathbf{x}) = \frac{m}{2}\|\mathbf{x}\|^2$ is a quadratic function. In the context of posterior sampling, the associated task is sampling from a distribution with a Gaussian prior. The composite structure also includes the general $m$-strongly convex function, which can be divided into $g(\mathbf{x}) = \frac{m}{2}\|\mathbf{x}\|^2$ and the weakly convex function $f(\mathbf{x}) = U(\mathbf{x}) - \frac{m}{2}\|\mathbf{x}\|^2$. We make the following assumption on $f$.

**Assumption 3.1.** $f \in \mathcal{C}^2$ is convex and has $L$-Lipschitz continuous gradients, i.e. $\mathbf{0} \preceq \nabla^2 f \preceq LI$.

It corresponds to making $m$-strongly convex and $L + m$-Lipschitz smooth assumptions on the potential function $U$, which is a basic setting and widely studied in the Langevin sampling literature (see e.g. Dalalyan [2017], Cheng and Bartlett [2018], Cheng et al. [2018], Shen and Lee [2019]).

In addition to the previous assumption, we follow Freund et al. [2022] to characterize the convergence rate by a new factor $H$ to avoid explicit dimension dependence.

**Definition 3.2.** Let $H$ be an upper bound of the Hessian matrices of $f$, i.e. $H \succeq \nabla^2 f(\mathbf{x})$.

Note that the Lipschitz smooth condition in Assumption 3.1 provides a loose bound for $H$ since we always have $H \preceq LI$. This implies that $\text{tr}(H)$ will reach $dL$ in the worst case, whereas this quantity can be much smaller than $dL$ under wide conditions. One typical example is when $f$ admits a so-called ridge separable structure shown below.

### 3.3  Example: ridge separable functions

**Definition 3.3.** $f$ is said to admit the ridge separable form if

$$f(\mathbf{x}) = \frac{1}{n}\sum_{i=1}^{n}\sigma_i(\mathbf{a}_i^T\mathbf{x}), \tag{3.2}$$

where $\sigma_i$ are all univariate functions, and $\mathbf{a}_i$ are given vectors in $\mathbb{R}^d$.

Ridge separable functions contain many applications in machine learning, such as regression or classification over generalized linear models as well as (deep) neural networks in the neural tangent kernel regime [Gelman and Hill, 2006, Jacot et al., 2018]. We follow the argument of Freund et al. [2022], showing realizable conditions that ensure $\mathrm{tr}(H)$ to be dimension-free.

**Assumption 3.4.** The function $\sigma_i \in \mathcal{C}^2$ has a bounded second derivative, i.e. $\sigma_i'' \leq L_0$ for all $i \in [n]$.

**Assumption 3.5.** For all $i \in [n]$, then norm of $\mathbf{a}_i$ is bounded by $R$, i.e. $\|\mathbf{a}_i\|^2 \leq R^2$.

When Assumptions 3.4 and 3.5 hold, the Hessian has the upper bound $\nabla^2 f(\mathbf{x}) = \sum_{i=1}^n \frac{1}{n} \sigma''(\mathbf{a}_i^T \mathbf{x}) \mathbf{a}_i \mathbf{a}_i^T \preceq \frac{L_0}{n} \sum_{i=1}^n \mathbf{a}_i \mathbf{a}_i^T$. Let $H = \frac{L_0}{n} \sum_{i=1}^n \mathbf{a}_i \mathbf{a}_i^T$ and

$$\mathrm{tr}(H) = \mathrm{tr}\left(\sum_{i=1}^n \frac{L_0}{n} \mathbf{a}_i \mathbf{a}_i^T\right) = \frac{L_0}{n} \sum_{i=1}^n \|\mathbf{a}_i\|^2 \leq L_0 R^2,$$

thus illustrates that $\mathrm{tr}(H)$ has a dimension-free $L_0 R^2$ upper bound. Meanwhile the Lipschitz constant of $\nabla f$ can be bounded by

$$\mathrm{Lip}(\nabla f) \leq \|H\|_2 = \left\| \frac{L_0}{n} \sum_{i=1}^n \mathbf{a}_i \mathbf{a}_i^T \right\| \leq L_0 R^2.$$

It indicates that worst-case upper bounds of $\mathrm{tr}(H)$ and $\mathrm{Lip}(\nabla f)$ are the same.

Note that Assumptions 3.4 and 3.5 are easy to achieve in practice. For example, consider using posterior sampling to compute the Bayes estimator over a linear model, whose advantages against maximum a posterior estimator have been discussed broadly (see e.g. Audibert [2009]). Here, $\mathbf{a}_i$ is associated with the data. So Assumption 3.5 can be realized by simply normalizing the data. Moreover, $\sigma_i$ corresponds to the loss function and is only required to have a bounded second derivative by Assumption 3.4. Note that sampling from a ridge separable potential functions are extensively studied in related literature (see [Mou et al., 2021, Vono et al., 2022, Lee et al., 2018] as examples).

### 3.4 Preliminary

**Overdamped Langevin.** The overdamped Langevin dynamics for target distribution (3.1) is a diffusion process that evolves along the following SDE

$$\mathrm{d}\mathbf{x}(t) = -\nabla g(\mathbf{x}(t))\mathrm{d}t - \nabla f(\mathbf{x}(t))\mathrm{d}t + \sqrt{2}\mathrm{d}\mathbf{B}_t \tag{3.3}$$

where $\mathbf{B}_t$ is the standard Brownian motion. The overdamped Langevin algorithms simulate and discretize the SDE (3.3). Different algorithms vary mainly by different discretization methods.

**Underdamped Langevin** Underdamped Langevin algorithms accelerate the convergence rate of overdamped Langevin algorithms and instead discretize the following dynamics

$$\begin{aligned} \mathrm{d}\mathbf{x}(t) &= \mathbf{v}(t)\mathrm{d}t, \\ \mathrm{d}\mathbf{v}(t) &= -u\nabla g(\mathbf{x}(t))\mathrm{d}t - u\nabla f(\mathbf{x}(t))\mathrm{d}t - 2\mathbf{v}(t)\mathrm{d}t + 2\sqrt{u}\mathrm{d}\mathbf{B}_t. \end{aligned} \tag{3.4}$$

The diffusion process has the stationary distribution $p_{(\mathbf{x}, \mathbf{v})} \propto \exp\left\{-\frac{1}{2u}\|\mathbf{v}\|^2 - U(\mathbf{x})\right\}$.

**Contractive Property** One notable property that yields the convergence of Langevin dynamics is the contractive property [Cheng and Bartlett, 2018, Cheng et al., 2018], given as follows.

**Definition 3.6.** A stochastic differential equation has contractive property if there exists a positive constant $m$ and

$$\mathbb{E}\|\mathbf{x}(t) - \mathbf{y}(t)\|^2 \leq \mathbb{E}\|\mathbf{x}(0) - \mathbf{y}(0)\|^2 \exp(-mt), \tag{3.5}$$

for any pair of solutions $\mathbf{x}(t)$ and $\mathbf{y}(t)$ driven by the same Brownian motion.

**Remark 3.7.** $\mathbb{E}\|\mathbf{x}(t) - \mathbf{y}(t)\|^2$ is an upper bound of squared Wasserstein distance of $\mathrm{Law}(\mathbf{x}(t))$ and $\mathrm{Law}(\mathbf{y}(t))$, and thus (3.5) implies a geometric convergence of the Wasserstein distance.

Contractive property can be established for both underdamped and overdamped Langevin dynamics under suitable conditions. The convergence analysis for our proposed algorithms follows a similar argument as the contractive property.

**Algorithm 1** Double-Randomized Underdamped Langevin (DRUL)

---

**Require:** Iteration $N$, target function $U = \frac{m}{2}\|\mathbf{x}\|^2 + f(\mathbf{x})$, initial point $(\mathbf{x}_0, \mathbf{v}_0)$, max step size $h$,
$u = \frac{1}{m+L}$ and $\kappa = \frac{L+m}{m}$.

    set $\rho(\mathrm{d}t) \propto \left(\frac{1}{h} - \frac{t}{h\kappa}\right)\mathrm{d}t$ with support $[0, h]$.

    set $\rho'(\mathrm{d}t) \propto \left(e^{\frac{t-h}{\kappa}} - \frac{t}{h}\right)\mathrm{d}t$ with support $[0, h]$.

    **for** $n = 1, 2, \cdots, N$ **do**

        Sample $\alpha_n \sim \rho'$ and $\beta_n \sim \rho$.

        Obtain the covariance matrix $\Sigma(\alpha_n, \beta_n)$.

        Sample random vector $(G, H, W)$ from distribution $\mathcal{N}(0, \Sigma)$.

        $\widehat{\mathbf{x}}_n \leftarrow A_{11}(\alpha_n)\mathbf{x}_n + A_{12}(\alpha_n)\mathbf{v}_n + u\int_0^{\alpha_n} A_{12}(s - \alpha_n)\nabla f(\mathbf{x}_n)\mathrm{d}s + 2\sqrt{u}H$.

        Update $\mathbf{x}_{n+1} \leftarrow A_{11}(\beta_n)\mathbf{x}_n + A_{12}(\beta_n)\mathbf{v}_n + u\int_0^{\beta_n} A_{12}(s - \beta_n)\nabla f(\widehat{\mathbf{x}}_n)\mathrm{d}s + 2\sqrt{u}G$.

        Update $\mathbf{v}_{n+1} \leftarrow A_{21}(\beta_n)\mathbf{x}_n + A_{22}(\beta_n)\mathbf{v}_n + u\int_0^{\beta_n} A_{22}(s - \beta_n)\nabla f(\widehat{\mathbf{x}}_n)\mathrm{d}s + 2\sqrt{u}W$.

    **end for**

---

## 4   Double-randomized underdamped Langevin algorithm

In this section, we introduce our double-randomized sampling method and present our main result. We will illustrate our intuition in Section 5.

The proposed algorithm is built upon the following discretization scheme given a fixed point $\widetilde{x}_n$

$$
\begin{aligned}
\mathrm{d}\mathbf{x}_n(t) &= \mathbf{v}_n(t)\mathrm{d}t, \\
\mathrm{d}\mathbf{v}_n(t) &= -u\nabla g(\mathbf{x}_n(t))\mathrm{d}t - u\nabla f(\widetilde{\mathbf{x}}_n)\mathrm{d}t - 2\mathbf{v}_n(t)\mathrm{d}t + 2\sqrt{u}\mathrm{d}\mathbf{B}_t.
\end{aligned}
\tag{4.1}
$$

The purpose of splitting the strongly convex part of $U(x)$ is to avoid the $md$ dimension dependence of $\mathrm{tr}(H)$. Although $m$ is reasonably small in practice, one cannot obtain a fully dimensional-free convergence rate if the term with respect to $g$ is discretized.

Denote the solution of process (4.1) at time $t$ given starting point $(\mathbf{x}_n, \mathbf{v}_n)$, Brownian motion $\{\mathbf{B}_t\}_{0 \leq t \leq \beta}$, step size $\beta$ and point $\widetilde{x}_n$ by $\mathcal{J}(\beta, \widetilde{x}_n; \{\mathbf{B}_t\}_{0 \leq t \leq \beta}, (\mathbf{x}_n, \mathbf{v}_n))$. The double-randomized algorithm performs the following one-step update:

$$
\mathbf{x}_{k+1} = \mathcal{J}(\beta, \mathcal{J}(\alpha, \mathbf{x}_n; \{\mathbf{B}_t\}_{0 \leq t \leq \alpha}, (\mathbf{x}_n, \mathbf{v}_n)); \{\mathbf{B}_t\}_{0 \leq t \leq \beta}, (\mathbf{x}_n, \mathbf{v}_n)).
\tag{4.2}
$$

The algorithm introduces a random step size $\beta \sim \rho \propto \left(\frac{1}{h} - \frac{t}{h\kappa}\right)$ defined on $[0, h]$ other than deterministic $h$. And let $\alpha$ follows the distribution $\rho' \propto \left(e^{\frac{t-h}{\kappa}} - \frac{t}{h}\right)$ on $[0, h]$.

The analysis focuses on a Gaussian prior setting. Under a Gaussian prior, the following Lemma points out that $\mathcal{J}$ is linear in starting point $(\mathbf{x}_n, \mathbf{v}_n)$ and has a decoupled Brownian motion, and thus the update (4.2) can be easily implemented.

**Lemma 4.1.** Assume $g(\mathbf{x}) = \frac{m}{2}\|\mathbf{x}\|^2$. If $\mathbf{x}_n(t)$ follows the discretized Langevin diffusion process (4.1) with starting point $(\mathbf{x}_n, \mathbf{v}_n)$ and a given $\widetilde{\mathbf{x}}_n$, for any $t > 0$, it satisfies the integral equation

$$
\begin{aligned}
\mathbf{x}_n(t) = \mathcal{J}(t, \widetilde{\mathbf{x}}_n; \{\mathbf{B}_s\}_{0 \leq s \leq t}, (\mathbf{x}_n, \mathbf{v}_n)) = A_{11}(t)\mathbf{x}_n + A_{12}(t)\mathbf{v}_n \\
+ u\int_0^t A_{12}(s - t)\nabla f(\widetilde{\mathbf{x}}_n)\mathrm{d}s + 2\sqrt{u}\int_0^t A_{12}(s - t)\mathrm{d}\mathbf{B}_s,
\end{aligned}
\tag{4.3}
$$

where

$$
\begin{aligned}
A_{11}(t) &= \frac{\sqrt{1 - um} - 1}{2\sqrt{1 - um}}e^{(-1 - \sqrt{1 - um})t} + \frac{\sqrt{1 - um} + 1}{2\sqrt{1 - um}}e^{(-1 + \sqrt{1 - um})t}, \\
A_{12}(t) &= -\frac{1}{2\sqrt{1 - um}}e^{(-1 - \sqrt{1 - um})t} + \frac{1}{2\sqrt{1 - um}}e^{(-1 + \sqrt{1 - um})t}
\end{aligned}
$$

are deterministic functions of $t$. Moreover, if $\mathbf{x}_n^*(t)$ follows the exact Langevin process (3.4) with starting point $(\mathbf{x}_n, \mathbf{v}_n)$, then for any $t > 0$

$$
\begin{aligned}
\mathbf{x}_n^*(t) = {}& A_{11}(t)\mathbf{x}_n + A_{12}(t)\mathbf{v}_n + u \int_0^t A_{12}(s-t)\nabla f(\mathbf{x}_n^*(s))\mathrm{d}s \\
& + 2\sqrt{u} \int_0^t A_{12}(s-t)\mathrm{d}\mathbf{B}_s.
\end{aligned}
\tag{4.4}
$$

Given the integral formula (4.3), the algorithm can be summarized as Algorithm 1. In Algorithm 1, $A_{21}$, $A_{22}$ is deterministic scalar functions, and $\Sigma : \mathbb{R}^2 \to \mathbb{R}^{3d \times 3d}$ is also deterministic. For the explicit formula, please refer to Appendix A. The distribution of $(H, G, W)$ is induced by the coupling between $\{\mathbf{B}_t\}_{0 \le t \le \alpha}$ and $\{\mathbf{B}_t\}_{0 \le t \le \beta}$ in (4.2).

## 4.1  Main theorem

The convergence guarantee of DRUL can be stated as the following theorem.

**Theorem 4.2** (**Main theorem**, convergence of DRUL). For any tolerance $\epsilon \in (0, 1)$, denote the minimizer of $U(\mathbf{x})$ by $\mathbf{x}_*$ and set the step size

$$
h \le \min \left\{ \frac{1}{12C_2\kappa}, \frac{\epsilon^{2/3}}{\left(24C_2\kappa(\frac{1}{m(L+m)}\mathrm{tr}(H) + \|\mathbf{x}_*\|^2)\right)^{1/3}} \right\},
$$

where $C_2 \ge 1$ is a universal constant. With initial point $(\mathbf{x}_0, \mathbf{v}_0)$, define $\Omega_0 = \mathbb{E}_{\mathbf{x} \sim p, \mathbf{v} \sim \mathcal{N}(0,u)} \left( \|\mathbf{x}_0 - \mathbf{x}\|^2 + \|\mathbf{x}_0 + \mathbf{v}_0 - \mathbf{v} - \mathbf{x}\|^2 \right)$. Then under Assumptions 3.1, when

$$
n \ge \frac{8e\kappa}{h} \log \left( \frac{2\mathbb{E}\Omega_0}{\epsilon^2} \right),
$$

Algorithm 1 outputs $\mathbf{x}_n$ such that $W_2(\mathrm{Law}(\mathbf{x}_n), p) \le \epsilon$.

Theorem 4.2 shows an overall $\widetilde{\mathcal{O}}\left( \frac{\left(\mathrm{tr}(H) + \|\mathbf{x}_*\|^2\right)^{1/3}}{\epsilon^{2/3}} \right)$ iteration and gradient complexity to find an $\epsilon$-approximate sample in 2-Wasserstein distance. By pre-finding $\mathbf{x}_*$ using a convex optimization algorithm and linearly drifting the coordinates, we can assume $\mathbf{x}_* = \mathbf{0}$ without loss of generality. DRUL admits the stepsize reaching $\mathcal{O}\left( \frac{\epsilon^{2/3}}{(\mathrm{tr}(H))^{1/3}} \right)$ and the overall complexity is $\widetilde{\mathcal{O}}\left( \frac{(\mathrm{tr}(H))^{1/3}}{\epsilon^{2/3}} \right)$.

Now we consider a realizable case where our result achieves better complexity on $d$. The concrete example is that $f$ admits a separable structure as discussed in Section 3.3. Corollary 4.3 below shows that when $f$ admits a ridge separable structure, Algorithm 1 enjoys a dimension-free iteration and gradient complexity.

**Corollary 4.3.** Follow the notations in Theorem 4.2. Further, if $f(\mathbf{x})$ admits a ridge-separable form and satisfies Assumptions 3.4 and 3.5. Then if we set the step size

$$
h \le \min \left\{ \frac{1}{12C_2\kappa}, \frac{\epsilon^{2/3}}{\left(24C_2\kappa(\frac{1}{m} + \|\mathbf{x}_*\|^2)\right)^{1/3}} \right\},
$$

With initial point $(\mathbf{x}_0, \mathbf{v}_0)$ and $\Omega_0 = \mathbb{E}_{\mathbf{x} \sim p, \mathbf{v} \sim \mathcal{N}(0,u)} \left( \|\mathbf{x}_0 - \mathbf{x}\|^2 + \|\mathbf{x}_0 + \mathbf{v}_0 - \mathbf{v} - \mathbf{x}\|^2 \right)$. Then when

$$
n \ge \frac{8e\kappa}{h} \log \left( \frac{2\mathbb{E}\Omega_0}{\epsilon^2} \right),
$$

Algorithm 1 outputs $\mathbf{x}_n$ such that $W_2(\mathrm{Law}(\mathbf{x}_n), p) \le \epsilon$.

**Discussion.** Theorem 4.2 establishes the convergence rate of DRUL with a weak dimension dependence. We shall note that for lots of high-dimensional sampling problems, the trace of the Hessian matrices for the potential function is much smaller than $d$ times the largest eigenvalue because the eigenvalues often drop rapidly. In practice, this situation has been considered in lots of topics. For

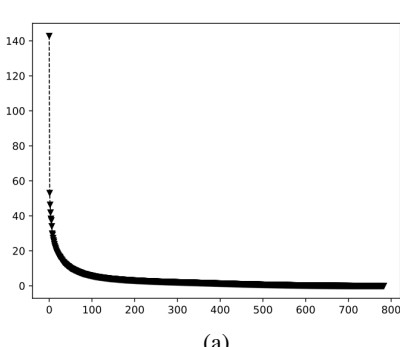
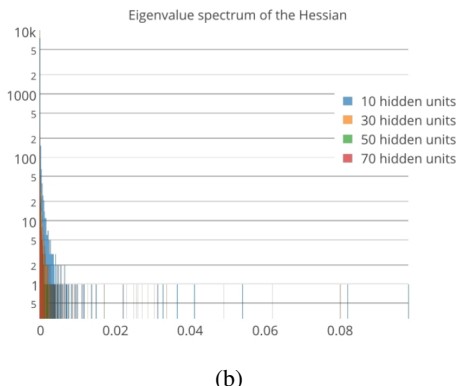

Figure 1: A demonstration of the eigenvalues of the Hessian matrix. (a) The eigenvalues of the Gram matrix of MNIST data. (b) Eigenvalues of a three-layer neural network from Sagun et al. [2016].

example, one defines the effective rank (e.g. Hsu et al. [2012]) to represent the acting dimension of the data on linear models. In distributed machine learning, one can show fewer bits are needed to be transmitted between machines when the eigenvalues decrease fast (e.g. Hanzely et al. [2018]). As shown in Figure 1(a), the Gram matrix of the MNIST dataset, which is also the Hessian of Bayesian ridge linear regression, has rapidly decreasing eigenvalues. Similar empirical results are observed on deep neural network models by Sagun et al. [2016], as in Figure 1(b). In fact, beyond the ridge separable case, there are many problems admitting $\mathrm{tr}(H) = o(d)$, such as the concentrated posterior distributions with bounded gradients and neural networks with regularization. However, one may notice that the result of Theorem 4.2 is based on a uniform upper bound of the Hessian matrices. We think this is the basic case to show that our algorithm achieves convergence with an intrinsically low-dimension dependence. It is possible to relax the condition to the upper bound of the traces for local Hessian matrices with an additional Hessian smoothness assumption. Please see more discussions in Appendix E.

## 5 Intuition

In this section, we provide the intuitions of the algorithm design. For the simplicity of the notation, let $\widehat{\mathbf{x}}_n(s) = \mathcal{J}(s, \mathbf{x}_n; \{\mathbf{B}_t\}_{0 \leq t \leq s}, (\mathbf{x}_n, \mathbf{v}_n))$ and $\mathbf{x}_n(s) = \mathcal{J}(s, \mathcal{J}(\alpha, \mathbf{x}_n; \{\mathbf{B}_t\}_{0 \leq t \leq \alpha}, (\mathbf{x}_n, \mathbf{v}_n)); \{\mathbf{B}_t\}_{0 \leq t \leq s}, (\mathbf{x}_n, \mathbf{v}_n))$. Then given $\alpha$ and $\beta$ in Algorithm 1, $\mathbf{x}_n(s) = \mathcal{J}(s, \widehat{\mathbf{x}}_n(\alpha); \{\mathbf{B}_t\}_{0 \leq t \leq \beta}, (\mathbf{x}_n, \mathbf{v}_n))$ and $\mathbf{x}_n(\beta) = \mathbf{x}_{n+1}$.

Our analysis tracks the dynamics of the distance to the stationary distribution. Specifically, we analysis the dynamics of $\mathbb{E}\Omega_n(t) = \mathbb{E}\|\mathbf{x}_n(t) - \mathbf{x}_n^*(t) + \mathbf{v}_n(t) - \mathbf{v}_n^*(t)\|^2 + \mathbb{E}\|\mathbf{x}_n(t) - \mathbf{x}_n^*(t)\|^2$ and $\mathbb{E}\Omega_n = \mathbb{E}\Omega_n(0)$, which upper bound the squared 2-Wasserstein distance $W_2(\mathrm{Law}(\mathbf{x}_n(t)), p)^2$ and $W_2(\mathrm{Law}(\mathbf{x}_n), p)^2$, respectively. Here, $(\mathbf{x}_n^*(t), \mathbf{v}_n^*(t))$ evolving along the exact Langevin diffusion (3.4) follows the stationary distribution and is synchronously coupled with $(\mathbf{x}_n(t), \mathbf{v}_n(t))$. One can find its formal definition in Appendix B. Via tracking the flow and using the contraction property of the process, the following Lemma characterizes the one-step discretization.

**Lemma 5.1.** Let $\mathbf{x}_n$ be the $n$-step output of Algorithm 1 and $\mathbf{x}_n^*(t)$ be defined as above. Set $u = \frac{1}{L+m}$. Under Assumptions 3.1, given that $\mathbf{x}_n$ and $\mathbf{x}_n^*$ are coupled synchronously, we have for any $h > 0$

$$\mathbb{E}\Omega_{n+1} = \mathbb{E}\mathbb{E}_{\beta \sim \rho}\Omega_n(\beta) \leq \mathbb{E}_{\beta \sim \rho}e^{-\frac{\beta}{\kappa}}\mathbb{E}\Omega_n + \mathcal{E} \tag{5.1}$$

where $\mathcal{E}$ is

$$\mathcal{E} = 2u\mathbb{E}\mathbb{E}_{\alpha \sim \rho'}\mathbb{E}_{\beta \sim \rho}\int_0^\beta e^{\frac{s-\beta}{\kappa}}\langle \mathbf{x}_n(s) - \mathbf{x}_n^*(s) + \mathbf{v}_n(s) - \mathbf{v}_n^*(s), \nabla f(\mathbf{x}_n(s)) - \nabla f(\widehat{\mathbf{x}}_n(\alpha))\rangle \mathrm{d}s.$$

Lemma 5.1 indicates at each step, $\mathbb{E}\Omega_n$ contracts with a local discretization error. Telescoping (5.1) and upper bounding the term $\mathcal{E}$ yields the Theorem 4.2. To obtain the convergence rate in the

main theorem, we will show that Algorithm 1 guarantees an upper bound of $\mathcal{E}$ which (1) achieves the state-of-the-art dependence on stepsize and (2) satisfies that the dimension dependence can be controlled by the contraction.

**Improved discretization error.** We accomplish the first goal by matching the expectation, which is detailed by Lemma 5.2.

**Lemma 5.2.** Let $\rho$ be probability measure defined on $[0,h]$ satisfying $\rho(\mathrm{d}t) \propto \left(\frac{1}{h} - \frac{t}{h\kappa}\right)\mathrm{d}t$, and the probability measure $\rho'$ defined on $[0,h]$ satisfies $\rho'(\mathrm{d}t) \propto \left(e^{\frac{t-h}{\kappa}} - \frac{t}{h}\right)\mathrm{d}t$ on $[0,h]$. Then for measurable function $F(t)$, $\rho$ and $\rho'$ satisfy that

(A) There exists positive constant $C_1$ such that $\mathbb{E}_{t\sim\rho}\int_0^t e^{\frac{s-t}{\kappa}}F(t)\mathrm{d}s = C_1 h\mathbb{E}_{t\sim\rho'}F(t)$.

(B) There exists positive constant $C_2$ such that $\mathbb{E}_{t\sim\rho'}|F(t)| \leq C\mathbb{E}_{t\sim\rho}|F(t)|$.

Claim (A) in Lemma 5.2 states that by choosing a random step size $\beta \sim \rho'$, we can leverage the low discretization error of the randomized midpoint method. Denote the random weight $\mathbf{w}_n(s,\alpha) = \mathbf{x}_n(s) - \mathbf{x}_n^*(s) + \mathbf{v}_n(s) - \mathbf{v}_n^*(s)$. One can split $\mathbf{w}_n(s,\alpha)$ into $(\mathbf{w}_n(s,\alpha) - \mathbf{w}_n(0,0)) + \mathbf{w}_n(0,0)$. The former term can be bounded using the one-step move, and by claim (A), the dominating latter one is

$$2u\mathbb{E}\mathbb{E}_{\alpha\sim\rho'}\mathbb{E}_{\beta\sim\rho}\int_0^\beta e^{\frac{s-\beta}{\kappa}}\langle\mathbf{w}(0,0), \nabla f(\mathbf{x}_n(s)) - \nabla f(\widehat{\mathbf{x}}_n(\alpha))\rangle\mathrm{d}s$$

$$=2uC_1 h\langle\mathbf{w}(0,0), \mathbb{E}\mathbb{E}_{s\sim\rho'}\left(\nabla f(\mathbf{x}_n(s)) - \nabla f(\widehat{\mathbf{x}}_n(s))\right)\rangle,$$

which attains a low discretization error given that $\mathbb{E}_{s\sim\rho'}\nabla f(\widehat{\mathbf{x}}_n(s))$ is a low biased approximation to $\mathbb{E}_{s\sim\rho'}\nabla f(\mathbf{x}_n(s))$.

**Averaged contraction can control the weight variance.** Then we consider the variation of the weight $w_n(s,\alpha)$. The time difference of $\mathbf{w}_n(s,\alpha)$ writes

$$\mathbf{w}(s,\alpha) - (\mathbf{x}_n - \mathbf{x}_n^* + \mathbf{v}_n - \mathbf{v}_n^*) = \int_0^s (\mathbf{v}_n(r) - \mathbf{v}_n^*(r) - um(\mathbf{x}_n(r) - \mathbf{x}_n^*(r)) \tag{5.2}$$
$$- 2(\mathbf{v}_n(r) - \mathbf{v}_n^*(r)) - u\nabla f(\widehat{\mathbf{x}}_n(\alpha)) + u\nabla f(\mathbf{x}_n^*(r)))\mathrm{d}r.$$

(5.2) indicates the variance $\mathrm{Var}_{s\sim\mu}(\mathbf{w}(s,\alpha))$ is dimension dependent for nondegenerated distributions $\mu$ on $[0,h]$, since it will introduce the $\mathbf{v}_n(r)$ whose difference to $\mathbf{v}_n(\beta)$ or $\mathbf{v}_n(0)$ is dimension dependent. We control the dimension dependence via the averaged contraction. And the randomized step size makes it possible to consider the averaged effect. We have the following Lemma to bound the variation of the weight.

**Lemma 5.3.** Let $\mathbf{x}_n(t), \mathbf{v}_n(t), \mathbf{v}_n^*(t)$ and $\mathbf{v}_n^*(t)$ be defined as above and $u = \frac{1}{L+m}$. Under Assumption 3.1, for any $t, \alpha \leq h$, we have

$$\mathbb{E}\|\mathbf{x}_n(t) - \mathbf{x}_n - \mathbf{x}_n^*(t) + \mathbf{x}_n^* + \mathbf{v}_n(t) - \mathbf{v}_n - \mathbf{v}_n^*(t) + \mathbf{v}_n^*\|^2$$
$$\lesssim h^2\mathbb{E}\mathbb{E}_{t\sim\rho}\Omega_n(t) + h^4\mathbb{E}\Omega_n + u^2h^4\frac{L}{m}\mathrm{tr}(H) + h^4\|\mathbf{x}_*\|^2. \tag{5.3}$$

Now the dimension dependence of $\mathrm{Var}(\mathbf{w}(s,\alpha))$ is contained in $\mathbb{E}_{t\sim\rho}\Omega_n(t)$, which can be controlled using the averaged contraction under the stochastic step size.

## 6  Conclusion

This paper proposes a double-randomized technique and designs the DRUL algorithm. We prove that with strongly convex and Lipschitz smooth assumptions potentials, the algorithm converges to the target distribution in Wasserstein distance in $\widetilde{\mathcal{O}}\left(\frac{(\mathrm{tr}(H)+\|\mathbf{x}_*\|^2)^{1/3}}{\epsilon^{2/3}}\right)$ iterations. The result illustrates that many sampling tasks in machine learning can achieve a dimension-independent complexity. The proposed DRUL algorithm can be potentially much faster than existing algorithms for high-dimensional problems. As a concrete example, when the negative log-likelihood function admits a ridge-separable structure, under mild conditions, a dimension-free $\widetilde{\mathcal{O}}\left(\frac{1}{\epsilon^{2/3}}\right)$ iteration complexities can be obtained by DRUL. We hope our technique brings new insights for designing dimension-independent algorithms for high-dimensional sampling.

## 7 Acknowledgement

C. Fang was supported by National Key R&D Program of China (2022ZD0114902), the NSF China (No. 62376008) and Wudao Foundation. T. Zhang was supported by the General Research Fund (GRF) of Hong Kong (No. 16310222).

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

# A  Simulation of the discretization process

## A.1  Integral form of the discretized process

### Proof of Lemma 4.1

*Proof.* Define the aligned vector $\mathbf{z}_n(s) = (\mathbf{x}_n(s), \mathbf{v}_n(s))$ and write the diffusion process (4.1) as

$$\mathrm{d}\mathbf{z}_n(s) = A\mathbf{z}_n(s)\mathrm{d}s + ua(\widehat{\mathbf{x}}_n)\mathrm{d}s + 2\sqrt{u}B\mathrm{d}\widetilde{\mathbf{B}}_s, \qquad (A.1)$$

where

$$A = \begin{bmatrix} 0 & I \\ -umI & -2I \end{bmatrix}, a = \begin{bmatrix} 0 \\ -\nabla f(\widehat{\mathbf{x}}) \end{bmatrix}, B = \begin{bmatrix} 0 & 0 \\ 0 & I \end{bmatrix}.$$

And $\widetilde{\mathbf{B}}_s$ is a Brownian Motion in $\mathbb{R}^{2d}$. By a variation of constant method, the solution of (A.1) is

$$\mathbf{z}_n(t) = e^{At}\mathbf{z}_n(0) + u\left(\int_0^t e^{-A(s-t)}\mathrm{d}s\right)a + 2\sqrt{u}\int_0^t e^{-A(s-t)}B\mathrm{d}\widetilde{\mathbf{B}}_s$$

To obtain the explicit expression of $\mathbf{x}_n(t)$ and $\mathbf{v}_n(t)$, respectively, it suffices to present a block-wise form

$$e^{At} = \begin{bmatrix} A_{11}(t)I & A_{12}(t)I \\ A_{21}(t)I & A_{22}(t)I \end{bmatrix}$$

of the exponential $e^{At}$. And a direct decomposition yields that

$$\begin{aligned}
A_{11}(t) &= \frac{\sqrt{2^2 - 4um} - 2}{2\sqrt{2^2 - 4um}}e^{(-2-\sqrt{2^2-4um})t/2} + \frac{\sqrt{2^2 - 4um} + 2}{2\sqrt{2^2 - 4um}}e^{(-2+\sqrt{2^2-4um})t/2}, \\
A_{12}(t) &= -\frac{1}{\sqrt{2^2 - 4um}}e^{(-2-\sqrt{2^2-4um})t/2} + \frac{1}{\sqrt{2^2 - 4um}}e^{(-2+\sqrt{2^2-4um})t/2}, \\
A_{21}(t) &= \frac{um}{\sqrt{2^2 - 4um}}e^{(-2-\sqrt{2^2-4um})t/2} + \frac{-um}{\sqrt{2^2 - 4um}}e^{(-2+\sqrt{2^2-4um})t/2}, \\
A_{22}(t) &= \frac{2 + \sqrt{2^2 - 4um}}{2\sqrt{2^2 - 4um}}e^{(-2-\sqrt{2^2-4um})t/2} + \frac{\sqrt{2^2 - 4um} - 2}{2\sqrt{2^2 - 4um}}e^{(-2+\sqrt{2^2-4um})t/2}.
\end{aligned} \qquad (A.2)$$

Since $\mathbf{z}_n(t) = (\mathbf{x}_n(t), \mathbf{v}_n(t))$, we obtain the close-form solution of the discretized Langevin process of $\mathbf{x}_n(t)$

$$\mathbf{x}_n(t) = A_{11}(t)\mathbf{x}_n + A_{12}(t)\mathbf{v}_n + u\int_0^h A_{12}(s-t)\nabla f(\widehat{\mathbf{x}}_n)\mathrm{d}s + 2\sqrt{u}\int_0^t A_{12}(s-t)\mathrm{d}\mathbf{B}_s.$$

A similar proof can be applied to the exact diffusion process and thus $x_n^*(t)$ satisfies

$$\mathbf{x}_n^*(t) = A_{11}(t)\mathbf{x}_n + A_{12}(t)\mathbf{v}_n + u\int_0^t A_{12}(s-t)\nabla f(\mathbf{x}^*(s))\mathrm{d}s + 2\sqrt{u}\int_0^t A_{12}(s-t)\mathrm{d}\mathbf{B}_s.$$

Thus we can obtain the result in Lemma 4.1. $\qquad\square$

## A.2  Controls of the integral forms

**Lemma A.1.** The functions $A_{11}(t)$ and $A_{12}(t)$ in Lemma 4.1 satisfy

$$|A_{12}(t)| \lesssim t, \qquad |A_{11}(t) - 1| \lesssim t$$

for $t \in [0, h]$.

*Proof.* In the proof, we will repeatedly use the inequality $e^t - 1 = \mathcal{O}(t)$. For notational simplicity, denote $\sqrt{1 - um}$ by $k$. For the first claim,

$$A_{12}(t) = -\frac{1}{2k}e^{-t-kt} + \frac{1}{2k}e^{-t+kt} = e^{-t}\left(\frac{1 - e^{-kt}}{2k} + \frac{e^{kt} - 1}{2k}\right) = \mathcal{O}(t).$$

As for the second claim, we have

$$A_{12}(t) - 1 = \frac{k-1}{2k}e^{-t-kt} + \frac{k+1}{2k}e^{-t+kt} - 1$$

$$= e^{-t}\left(\frac{e^{-kt} + e^{kt} - 2}{2} + \frac{1 - e^{-kt}}{2k} + \frac{e^{kt} - 1}{2k}\right) = \mathcal{O}(kt) + \mathcal{O}(t).$$

Note that $k = \sqrt{1 - um} \leq 1$ and $h \leq \frac{1}{\kappa} \leq 1$. When $t \leq h$ our claims hold true. $\square$

### A.3 Simulation of the Brownian motion

**Lemma A.2.** Let $\mathbf{B}_t$ be a standard Brownian Motion. In Algorithm 1, we have $G = \int_0^t A_{12}(s - t)\mathrm{d}\mathbf{B}_s$, $H = \int_0^\alpha A_{12}(s - \alpha)\mathrm{d}\mathbf{B}_s$ and $W = \int_0^t A_{22}(s - t)\mathrm{d}\mathbf{B}_s$. Then conditioned on the randomness of $\alpha$ and $\beta$, $(G, H, W)$ follows a mean-zero Gaussian distribution with covariance

$$\mathbb{E}[(G - \mathbb{E}G)(H - \mathbb{E}H)^T] = \left(\int_0^{\min(t,\alpha)} A_{12}(s - t)A_{12}(s - \alpha)\mathrm{d}s\right)I,$$

$$\mathbb{E}[(G - \mathbb{E}G)(G - \mathbb{E}G)^T] = \left(\int_0^t A_{12}(s - t)^2\mathrm{d}s\right)I,$$

$$\mathbb{E}[(H - \mathbb{E}H)(H - \mathbb{E}H)^T] = \left(\int_0^\alpha A_{12}(s - \alpha)^2\mathrm{d}s\right)I,$$

$$\mathbb{E}[(W - \mathbb{E}W)(W - \mathbb{E}W)^T] = \left(\int_0^t A_{22}(s - t)^2\mathrm{d}s\right)I,$$

$$\mathbb{E}[(W - \mathbb{E}W)(H - \mathbb{E}H)^T] = \left(\int_0^{\min(\alpha,t)} A_{12}(s - \alpha)A_{22}(s - t)\mathrm{d}s\right)I,$$

$$\mathbb{E}[(W - \mathbb{E}W)(G - \mathbb{E}G)^T] = \left(\int_0^t A_{12}(s - t)A_{22}(s - t)\mathrm{d}s\right)I.$$

(A.3)

*Proof.* By the properties of Brownian motion, $(G, H, W)$ is a zero-mean Gaussian variable. Its variance is given by

$$\mathbb{E}[(G - \mathbb{E}G)(H - \mathbb{E}H)^T] = \mathbb{E}\left[\int_0^t A_{12}(s - t)\mathrm{d}\mathbf{B}_s \int_0^\alpha A_{12}(s - \alpha)\mathrm{d}\mathbf{B}_s\right]$$

$$= \left(\int_0^{\min(t,\alpha)} A_{12}(s - t)A_{12}(s - \alpha)\mathrm{d}s\right)I.$$

The rest of the claims can be proved similarly. $\square$

## B  Proof of the main theorem

In this section, we present the proof of Theorem 4.2.

### B.1  Contractive lemma

We begin by defining the n-step exact process $(\mathbf{x}_n^*, \mathbf{v}_n^*)$ and $(\mathbf{x}_n^*(t), \mathbf{v}_n^*(t))$. Denote $\mathrm{d}p^* \propto e^{-U(\mathbf{x}) - \frac{1}{2u}\|\mathbf{v}\|^2}\mathrm{d}\mathbf{x}\mathrm{d}\mathbf{v}$. Let $(\mathbf{x}_0^*, \mathbf{v}_0^*) \sim p^*$ and

$$\mathrm{d}\mathbf{x}_n^*(t) = \mathbf{v}_n^*(t)\mathrm{d}t, \qquad \mathrm{d}\mathbf{v}_n^*(t) = -uU(\mathbf{x}_n^*(t))\mathrm{d}t - 2\mathbf{v}_n^*(t)\mathrm{d}t + \sqrt{2u}\mathrm{d}\mathbf{B}_t,$$
$$(\mathbf{x}_n^*(0), \mathbf{v}_n^*(0)) = (\mathbf{x}_{n-1}^*(\beta_{n-1}), \mathbf{x}_{n-1}^*(\beta_{n-1})),$$

(B.1)

For the simplicity of notation, we denote $(\mathbf{x}_n^*, \mathbf{v}_n^*) = (\mathbf{x}_n^*(0), \mathbf{v}_n^*(0))$. And we have for any $n$ and $t$, $(\mathbf{x}_n^*(t), \mathbf{v}_n^*(t)) \sim p^*$. With slight abuse of notation, let $\widehat{\mathbf{x}}_n(t)$ and $\mathbf{x}_n(t)$ denote inaccurate starting point discretization process and midpoint process in Algorithm 1, respectively. And recall that in Algorithm 1 we have

$$\widehat{\mathbf{x}}_n(t) = A_{11}(t)\mathbf{x}_n + A_{12}(t)\mathbf{v}_n + u\int_0^t A_{12}(s-t)\nabla f(\mathbf{x}_n)\mathrm{d}s + 2\sqrt{u}\int_0^t A_{12}(s-t)\mathrm{d}\mathbf{B}_s \quad \text{(B.2)}$$

and

$$\mathbf{x}_n(t) = A_{11}(t)\mathbf{x}_n + A_{12}(t)\mathbf{v}_n + u\int_0^t A_{12}(s-t)\nabla f(\widehat{\mathbf{x}}_n(\alpha))\mathrm{d}s + 2\sqrt{u}\int_0^t A_{12}(s-t)\mathrm{d}\mathbf{B}_s. \tag{B.3}$$

And we assume $\widehat{\mathbf{x}}_n(t)$ is driven by the same Brownian Motion as $\mathbf{x}_n(t)$ and $\mathbf{x}_n^*(t)$ at the $n$-th step.

Define $\Omega_n(t) = \|\mathbf{x}_n(t) - \mathbf{x}_n^*(t)\|^2 + \|\mathbf{x}_n(t) - \mathbf{x}_n^*(t) + \mathbf{v}_n(t) - \mathbf{v}_n^*(t)\|^2$ and $\Omega_n = \|\mathbf{x}_n - \mathbf{x}_n^*\|^2 + \|\mathbf{x}_n - \mathbf{x}_n^* + \mathbf{v}_n - \mathbf{v}_n^*\|^2$. The proof concentrates on the dynamics of the $\Omega_n$. Note that Wasserstein distance minimizes all the couplings and $\Omega_n(t) \geq \|\mathbf{x}_n(t) - \mathbf{x}_n^*(t)\|^2$. Hence $\mathbb{E}\Omega_n$ upper bounds of the Wasserstein distance $W_2(\text{Law}(\mathbf{x}_n), p)$. Given a synchronously coupled assumption, the dynamics of $\Omega_n(t)$ can be characterized as follow.

### B.1.1   Proof of Lemma 5.1

*Proof.* Let the random process $\mathbf{B}_t^\alpha$ given $\mathbf{B}_\alpha$ be

$$\mathrm{d}\mathbf{B}_t^\alpha = \begin{cases} \frac{\mathbf{B}_\alpha - \mathbf{B}_t^\alpha}{\alpha - t} + \mathrm{d}\mathbf{B}_t', & 0 \leq t \leq \alpha \\ \mathrm{d}\mathbf{B}_t, & t \geq \alpha \end{cases}$$

where $\mathbf{B}_t'$ is an independent Brownian motion. Note that $\mathbf{B}_t^\alpha$ is a Brownian bridge and $\mathbf{B}_\alpha^\alpha = \mathbf{B}_\alpha$. Then when $\widehat{\mathbf{x}}_n(\alpha)$ is given and $\mathbf{x}_n^*(t), \mathbf{x}_n(t)$ are coupled synchronously, given $\mathbf{B}_\alpha$, we have

$$\mathrm{d}\mathbf{x}_n^*(t) = \mathbf{v}_n^*(t)\mathrm{d}t,$$
$$\mathrm{d}\mathbf{v}_n^*(t) = -u\nabla g(\mathbf{x}_n^*(t))\mathrm{d}t - u\nabla f(\mathbf{x}_n^*(t))\mathrm{d}t - 2\mathbf{v}_n^*(t)\mathrm{d}t + 2\sqrt{u}\mathrm{d}\mathbf{B}_t^\alpha,$$

and

$$\mathrm{d}\mathbf{x}_n(t) = \mathbf{v}_n(t)\mathrm{d}t,$$
$$\mathrm{d}\mathbf{v}_n(t) = -u\nabla g(\mathbf{x}_n(t))\mathrm{d}t - u\nabla f(\widehat{\mathbf{x}}_n(\alpha))\mathrm{d}t - 2\mathbf{v}_n(t)\mathrm{d}t + 2\sqrt{u}\mathrm{d}\mathbf{B}_t^\alpha.$$

For simplicity, denote $\mathbf{z}_n(t) = \mathbf{x}_n(t) - \mathbf{x}_n^*(t)$ and $\psi_n(t) = \mathbf{v}_n(t) - \mathbf{v}_n^*(t)$. Then by Taylor's Theorem

$$\nabla U(\mathbf{x}_n(t)) - \nabla U(\mathbf{x}_n^*(t)) = \int_0^1 \nabla^2 U(s\mathbf{x}_n(t) + (1-s)\mathbf{x}_n^*(t))\mathrm{d}s\mathbf{z}_n(t). \tag{B.4}$$

Define $\mathcal{H}_t = \int_0^1 \nabla^2 U(s\mathbf{x}_n(t) + (1-s)\mathbf{x}_n^*(t))\mathrm{d}s$, then

$$\mathrm{d}\Omega_n(t)/\mathrm{d}t = 2\Big\langle \mathbf{z}_n(t) + \psi_n(t), \psi_n(t) - u(g(\mathbf{x}_n(t)) - g(\mathbf{x}_n^*(t))) - u\Big(\nabla f(\widehat{\mathbf{x}}_n(\alpha)) - \nabla f(\mathbf{x}_n^*(t))\Big) - \gamma\psi_n(t)\Big\rangle$$
$$+ 2\Big\langle \mathbf{z}_n(t), \psi_n(t)\Big\rangle$$
$$= -2\Big(\underbrace{\langle \mathbf{z}_n(t) + \psi_n(t), (\gamma - 1)\psi_n(t) + u\mathcal{H}_t\mathbf{z}_n(t)\rangle - \langle \mathbf{z}_n(t), \psi_n(t)\rangle}_{A}\Big)$$
$$+ 2u\langle \mathbf{z}_n(t) + \psi_n(t), \nabla f(\mathbf{x}_n(t)) - \nabla f(\widehat{\mathbf{x}}_n(\alpha))\rangle,$$

where in the second equality we use (B.4) and divide $\nabla f(\widehat{\mathbf{x}}_n(\alpha))$ into $\nabla f(\mathbf{x}_n(t))$ and the difference $\nabla f(\widehat{\mathbf{x}}_n(\alpha)) - \nabla f(\mathbf{x}_n(t))$. Note that $A$ can be converted to a quadratic form

$$A = [\mathbf{z}_n(t)^T + \psi_n(t)^T, \mathbf{z}_n(t)^T]\begin{bmatrix} I & \frac{1}{2}u\mathcal{H}_t - I \\ \frac{1}{2}u\mathcal{H}_t - I & I \end{bmatrix}\begin{bmatrix} \mathbf{z}_n(t) + \psi_n(t) \\ \mathbf{z}_n(t) \end{bmatrix}$$

whose eigenvalues $\lambda_i$ are given by characteristic function $(1 - \lambda)^2 - \frac{1}{4}(u\lambda_i(\mathcal{H}_t) + um - 2)^2 = 0$. Given the strong-convexity and Lipschitz smoothness of the potential $U(x)$, when $u$ is set to be $\frac{1}{L+m}$ the eigenvalue of $A$ is greater than $\frac{1}{2\kappa}$. Therefore,

$$\mathrm{d}\Omega_n(t)/\mathrm{d}t \leq -\frac{1}{\kappa}\Omega_n(t) + 2u\langle \mathbf{z}_n(t) + \psi_n(t), \nabla f(\mathbf{x}_n(t)) - \nabla f(\widehat{\mathbf{x}}_n(\alpha)))\rangle.$$

Then by multiplying $e^{\frac{t}{\kappa}}$ and taking the integral form 0 to $\beta$, we obtain

$$\Omega_n(\beta) \leq e^{-\frac{\beta}{\kappa}}\Omega_n(0) + 2u\int_0^\beta e^{\frac{s-\beta}{\kappa}}\langle \mathbf{x}_n(s) - \mathbf{x}_n^*(s) + \mathbf{v}_n(s) - \mathbf{v}_n^*(s), \nabla f(\mathbf{x}_n(s)) - \nabla f(\widehat{\mathbf{x}}_n(\alpha))\rangle\mathrm{d}s.$$

In Algorithm 1 the random step size $\beta \sim \rho$ and thus we reach that

$$\mathbb{E}\Omega_{n+1} \leq \mathbb{E}_{\beta\sim\rho}e^{-\frac{\beta}{\kappa}}\mathbb{E}\Omega_n + 2u\mathbb{E}\mathbb{E}_{\beta\sim\rho}\int_0^\beta e^{\frac{s-\beta}{\kappa}}\langle \mathbf{x}_n(s) - \mathbf{x}_n^*(s) + \mathbf{v}_n(s) - \mathbf{v}_n^*(s), \nabla f(\mathbf{x}_n(s)) - \nabla f(\widehat{\mathbf{x}}_n(\alpha))\rangle\mathrm{d}s.$$

$\qquad\square$

## B.2 Proof of Theorem 4.2

We devote the rest of this section to the proof of the main theorem. Lemma 5.1 shows that the dynamics of the $\mathbb{E}\Omega_n$ is driven by a shrinkage term and a local discretization error. To achieve the claimed convergence rate, the proof aims to establish an upper bound of local error which is: (1) dimensional-independent, (2) optimal in the sense of max step size $h$.

*Proof.* Given that $\mathbb{E}\Omega_n$ is an upper bound of the squared 2-Wasserstein distance, it suffices to prove $\mathbb{E}\Omega_n \leq \epsilon^2$. By Lemma 5.1 and Lemma 5.2, we have

$$
\begin{aligned}
\mathbb{E}\Omega_n(\beta) \leq& \mathbb{E}_{\beta\sim\rho}e^{-\frac{\beta}{\kappa}}\mathbb{E}\Omega_n \\
&+ 2u\mathbb{E}\mathbb{E}_{\beta\sim\rho}\int_0^\beta e^{\frac{s-\beta}{\kappa}}\langle \mathbf{x}_n(s) - \mathbf{x}_n^*(s) + \mathbf{v}_n(s) - \mathbf{v}_n^*(s), \nabla f(\mathbf{x}_n(s)) - \nabla f(\widehat{\mathbf{x}}_n(\alpha))\rangle\mathrm{d}s \\
=& \mathbb{E}_{\beta\sim\rho}e^{-\frac{\beta}{\kappa}}\mathbb{E}\Omega_n \\
&+ \underbrace{2uhC_1\mathbb{E}\mathbb{E}_{\beta\sim\rho'}\langle \mathbf{x}_n(\beta) - \mathbf{x}_n^*(\beta) + \mathbf{v}_n(\beta) - \mathbf{v}_n^*(\beta), \nabla f(\mathbf{x}_n(\beta)) - \nabla f(\widehat{\mathbf{x}}_n(\alpha))\rangle}_{A}.
\end{aligned}
$$
$$\tag{B.5}$$

Next, we bound the error term $A$ by

$$
\begin{aligned}
A =& 2C_1uh\mathbb{E}\mathbb{E}_{\beta\sim\rho'}\langle \mathbf{x}_n(\beta) - \mathbf{x}_n - \mathbf{x}_n^*(\beta) + \mathbf{x}_n^* + \mathbf{v}_n(\beta) - \mathbf{v}_n - \mathbf{v}_n^*(\beta) + \mathbf{v}_n^*, \nabla f(\mathbf{x}_n(\beta)) - \nabla f(\widehat{\mathbf{x}}_n(\alpha))\rangle \\
&+ 2C_1uh\mathbb{E}\langle \mathbf{x}_n - \mathbf{x}_n^* + \mathbf{v}_n - \mathbf{v}_n^*, \mathbb{E}_{\alpha\sim\rho'}\mathbb{E}_{\beta\sim\rho'}(\nabla f(\mathbf{x}_n(\beta)) - \nabla f(\widehat{\mathbf{x}}_n(\alpha)))\rangle \\
\lesssim& \frac{uh}{uh}\mathbb{E}\|\mathbf{x}_n(\beta) - \mathbf{x}_n - \mathbf{x}_n^*(\beta) + \mathbf{x}_n^* + \mathbf{v}_n(\beta) - \mathbf{v}_n - \mathbf{v}_n^*(\beta) + \mathbf{v}_n^*\|^2 \\
&+ u^2h^2\mathbb{E}\|\nabla f(\mathbf{x}_n(\beta)) - \nabla f(\widehat{\mathbf{x}}_n(\alpha))\|^2 + \frac{h}{u}uh\mathbb{E}\|\mathbf{x}_n - \mathbf{x}_n^* + \mathbf{v}_n - \mathbf{v}_n^*\|^2 \\
&+ \frac{u}{h}uh\mathbb{E}\|\mathbb{E}_{\alpha\sim\rho'}\mathbb{E}_{\beta\sim\rho'}(\nabla f(\mathbf{x}_n(\beta)) - \nabla f(\widehat{\mathbf{x}}_n(\alpha)))\|^2 \\
\overset{a}{\lesssim}& \left(h^2\mathbb{E}_{\beta\sim\rho}\mathbb{E}\Omega_n(\beta) + h^4\mathbb{E}\Omega_n + \frac{u^2h^4L}{m}\mathrm{tr}(H) + h^4\|\mathbf{x}_*\|^2\right) \\
&+ u^2h^2\left(h^2L^2\mathbb{E}\Omega_n + \frac{h^2L}{m}\mathrm{tr}(H) + L^2h^2\|\mathbf{x}_*\|^2\right) + h^2\mathbb{E}\Omega_n \\
&+ u^2\left(h^4L^2\mathbb{E}\Omega_n + \frac{h^4L}{m}\mathrm{tr}(H) + h^4L^2\|\mathbf{x}_*\|^2\right) \\
\overset{b}{\lesssim}& h^2\mathbb{E}\mathbb{E}_{\beta\sim\rho}\Omega_n(\beta) + h^2\mathbb{E}\Omega_n + \frac{u^2h^4L}{m}\mathrm{tr}(H) + h^4\|\mathbf{x}_*\|^2,
\end{aligned}
$$

where in inequality $\overset{a}{\lesssim}$, we apply Lemma D.1, Lemma 5.3 and Lemma D.5; inequality $\overset{b}{\lesssim}$ follows by the observation $uL = \frac{L}{L+m} \leq 1$. Then by plugging the upper bound of $A$ into (B.5), there exists constant $C_2$ such that

$$\mathbb{E}\Omega_n(\beta) \leq \mathbb{E}_{\beta\sim\rho}e^{-\frac{\beta}{\kappa}}\mathbb{E}\Omega_n + C_2\left(h^2\mathbb{E}\Omega_n(\beta) + h^2\mathbb{E}\Omega_n + \frac{u^2h^4L}{m}\mathrm{tr}(H) + h^4\|\mathbf{x}_*\|^2\right). \quad (B.6)$$

(B.6) can be simplified as

$$\mathbb{E}\Omega_n(\beta) \leq \frac{\mathbb{E}_{\beta\sim\rho}e^{-\frac{\beta}{\kappa}} + C_2h^2}{1 - C_2h^2}\mathbb{E}\Omega_n + \frac{C_2h^4}{1 - C_2h^2}\left(\frac{u^2L}{m}\mathrm{tr}(H) + \|\mathbf{x}_*\|^2\right). \quad (B.7)$$

Note that when $\beta \sim \rho$, $\mathbb{E}\Omega_n(\beta) = \mathbb{E}\Omega_{n+1}$. And $\mathbb{E}\Omega_n$ is an upper bound of Wasserstein distance. Equation (B.7) characterizes the one-step discretization of the Langevin process, where the first term indicates a linear convergence if without discretization error, and the second term is the one-step discretization error.

Without loss of generality, let $C_2 \geq 1$. Since $h \leq 1 \leq \kappa$, we have $\mathbb{E}_{\beta\sim\rho}e^{-\frac{\beta}{\kappa}} = \frac{\kappa}{h}\left(1 - e^{\frac{h}{\kappa}}\right) \leq 1 - \frac{h}{3\kappa}$. Since $h \leq \frac{1}{12C_2\kappa}$, we have $h \leq \frac{1}{2C_2\kappa}$ and $h \leq \frac{1}{\sqrt{C_2}}$ and therefore $1 - C_2h^2 \geq 0$, $2C_2h^2 \leq \frac{h}{6\kappa}$. Hence the first term of (B.7) can be bounded by

$$\begin{aligned}
\frac{\mathbb{E}_{\beta\sim\rho}e^{-\frac{\beta}{\kappa}} + C_2h^2}{1 - C_2h^2}\mathbb{E}\Omega_n &\leq \frac{1 - \frac{h}{3\kappa} + C_2h^2}{1 - C_2h^2}\mathbb{E}\Omega_n \\
&= \left(\frac{-\frac{h}{3\kappa} + 2C_2h^2}{1 - C_2h^2} + 1\right)\mathbb{E}\Omega_n \\
&\leq \left(\frac{-\frac{h}{3\kappa} + \frac{h}{6\kappa}}{1 - C_2h^2} + 1\right)\mathbb{E}\Omega_n \\
&\leq \left(1 - \frac{h}{6\kappa}\right)\mathbb{E}\Omega_n,
\end{aligned} \quad (B.8)$$

As for error term in (B.7), since $h \leq \frac{1}{12C_2\kappa} \leq \frac{1}{\sqrt{2C_2}}$, we have $\frac{C_2h^4}{1 - C_2h^2} \leq 2C_2h^4$. Denote $r = 1 - \frac{h}{6\kappa}$ and $\mathcal{E} = 2C_2h^4\left(\frac{u^2L}{m}\mathrm{tr}(H) + \|\mathbf{x}_*\|^2\right)$ and therefore

$$\mathbb{E}\Omega_N \leq r\mathbb{E}\Omega_{N-1} + \mathcal{E} \leq r^N\mathbb{E}\Omega_0 + \mathcal{E}(1 + r + \cdots + r^{N-1}) \leq r^N\mathbb{E}\Omega_0 + \frac{\mathcal{E}}{1 - r}. \quad (B.9)$$

When $N \geq \frac{6}{\kappa}\log\left(\frac{2\mathbb{E}\Omega_0}{\epsilon^2}\right)$, we have

$$r^N\mathbb{E}\Omega_0 \leq e^{\frac{Nh}{6\kappa}}\mathbb{E}\Omega_0 \leq \frac{\epsilon^2}{2}. \quad (B.10)$$

And when $h \leq \frac{\epsilon^{2/3}}{\left(24C_2\kappa(\frac{u^2L}{m}\mathrm{tr}(H)+\|\mathbf{x}_*\|^2)\right)^{1/3}}$, we get

$$\frac{\mathcal{E}}{1 - r} \leq \frac{\epsilon^2}{2}. \quad (B.11)$$

Plugging (B.10) and (B.11) into (B.9) yields

$$\mathbb{E}\Omega_N \leq \epsilon^2,$$

and therefore completes the proof. □

# C  Supporting lemmas

We assume that Assumption 3.1 holds in this section and Section D.

## C.1 Proof of Lemma 5.2

We would like to point out that how to match the random step size $\alpha$ and $\rho$ is critical for obtaining a low-bias estimator. And the control for local discretization error in Lemma 5.1 involves a challenging multiple integral. These considerations inspire our choice of $\rho$ and $\rho'$. And in the following lemma, we summarize the main properties of $\rho$ and $\rho'$.

**Lemma C.1.** Let $\rho$ be probability measure defined on $[0, h]$ satisfying $\rho(\mathrm{d}t) \propto \left(\frac{1}{h} - \frac{t}{h\kappa}\right) \mathrm{d}t$, and the probability measure $\rho'$ defined on $[0, h]$ satisfies $\rho'(\mathrm{d}t) \propto \left(e^{\frac{t-h}{\kappa}} - \frac{t}{h}\right) \mathrm{d}t$ on $[0, h]$. Then for measurable function $F(t)$, $\rho$ and $\rho'$ satisfy that

(A) There exists positive constant $C_1$ such that $\mathbb{E}_{t\sim\rho} \int_0^t e^{\frac{s-t}{\kappa}} F(t) = C_1 h \mathbb{E}_{t\sim\rho'} F(t)$.

(B) There exists positive constant $C_2$ such that $\mathbb{E}_{t\sim\rho'} |F(t)| \leq C \mathbb{E}_{t\sim\rho} |F(t)|$.

*Proof.* We begin with the first claim. By the definition of $\rho$ and integration by parts, we obtain

$$
\begin{aligned}
\mathbb{E}_{t\sim\text{unif}[0,1]} \int_0^t e^{\frac{s-t}{\kappa}} F(s)\mathrm{d}s &= \int_0^h \frac{1}{h} \int_0^t e^{\frac{s-t}{\kappa}} F(s)\mathrm{d}s\mathrm{d}t \\
&= \left(\frac{t}{h} \int_0^t e^{\frac{s-t}{\kappa}} F(s)\mathrm{d}s\right)\Big|_0^h - \int_0^h \frac{t}{h}\left(F(t) - \frac{1}{\kappa} \int_0^t e^{\frac{s-t}{\kappa}} F(s)\mathrm{d}s\right) \mathrm{d}t \\
&= \int_0^h e^{\frac{t-h}{\kappa}} F(t)\mathrm{d}t - \int_0^h \frac{t}{h} F(t)\mathrm{d}t + \int_0^h \frac{t}{h\kappa} \int_0^t e^{\frac{s-t}{\kappa}} F(s)\mathrm{d}s\mathrm{d}t,
\end{aligned}
$$

(C.1)

which indicates that

$$
\int_0^h \left(\frac{1}{h} - \frac{t}{h\kappa}\right) \int_0^t e^{\frac{s-t}{\kappa}} F(s)\mathrm{d}s\mathrm{d}t = \int_0^h \left(e^{t-h}\kappa - \frac{t}{h}\right) F(t)\mathrm{d}t.
$$

$\rho \propto \frac{1}{h} - \frac{t}{h\kappa}$ has a normalizing constant which is of order $\Theta(1)$. Denote the normalizing constant of $\int_0^h \left(e^{\frac{t-h}{\kappa}} - \frac{t}{h}\right) \mathrm{d}s$ by a scalar function $Z'(h)$. To obtain the first claim, it suffices to show $Z'(h)$ is of order $\mathcal{O}(h)$. Then given that $e^{-\frac{h}{\kappa}} \geq 1 - \frac{h}{\kappa}$, $Z'(h)$ satisfies

$$
Z'(h) = \int_0^h \left(e^{\frac{t-h}{\kappa}} - \frac{t}{h}\right) \mathrm{d}t = \kappa\left(1 - e^{-\frac{h}{\kappa}}\right) - \frac{h}{2} \leq h - \frac{h}{2} = \frac{h}{2},
$$

which proves the first claim.

Then we prove the second claim of the proposition. Note that when $h \leq \kappa$

$$
Z'(h) = \kappa\left(1 - e^{-\frac{h}{\kappa}}\right) - \frac{h}{2} \geq \left(1 - \frac{1}{e}\right) h - \frac{h}{2} = \left(\frac{1}{2} - \frac{1}{e}\right) h.
$$

Thus

$$
\frac{1}{Z'(h)}\left(e^{\frac{t-h}{\kappa}} - \frac{t}{h}\right) \leq \frac{1}{Z'(h)} \lesssim \frac{1}{h}.
$$

(C.2)

As for $\rho$, denote the normalizing constant of $\rho$ by $Z(h) = \int_0^h \frac{1}{h} - \frac{t}{h\kappa}$. Since $h = o(1)$ and $\kappa > 1$,

$$
Z(h) = 1 - \frac{h}{2\kappa} = \Theta(1).
$$

Then we have

$$
\frac{1}{Z(h)}\left(\frac{1}{h} - \frac{t}{j\kappa}\right) \gtrsim \frac{1}{h}\left(1 - \frac{t}{\kappa}\right) \gtrsim \frac{1}{h},
$$

(C.3)

where the last inequality follows by that $t \leq h = 0(1)$ and $\kappa > 1$. Combining (C.2) and (C.3) yields Claim (B). $\qquad\square$

## C.2 Convergence lemma

**Lemma C.2.** Let $\mathbf{x}_*$ be the minimizer of the potential function, that is, $\mathbf{x}_* = \arg\min\{g(\mathbf{x}) + f(\mathbf{x})\}$, and $A$ is a positive definite matrix that $mI \preceq A \preceq LI$. Let $\mathbf{x}_n(t)$ and $\mathbf{x}_n^*(t)$ be defined as (B.3) and (B.1). Assume that $u = \frac{1}{L+m}$. Then we have the following bound

$$\mathbb{E}\|A\mathbf{x}_n(t)\|^2 \lesssim L^2\mathbb{E}\|\mathbf{x}_n(t) - \mathbf{x}_n^*(t)\|^2 + \frac{1}{m}\mathrm{tr}(A^2) + L^2\|\mathbf{x}_*\|^2,$$

$$\mathbb{E}\|A(\mathbf{x}_n(t) + \mathbf{v}_n(t))\|^2 \lesssim L^2\mathbb{E}\|\mathbf{x}_n(t) + \mathbf{v}_n(t) - \mathbf{x}_n^*(t) - \mathbf{v}_n^*(t)\|^2 + \frac{1}{m}\mathrm{tr}(A^2) + L^2\|\mathbf{x}_*\|^2,$$

and

$$\mathbb{E}\|A\nabla f(\mathbf{x}_n(t))\|^2 \lesssim L^4\mathbb{E}\|\mathbf{x}_n(t) - \mathbf{x}_n^*(t)\|^2 + \frac{L^2}{m}\mathrm{tr}(A^2) + L^2 m^2\|\mathbf{x}_*\|^2.$$

*Proof.* We begin with the first bound. By Young's inequality and Lipschitz smoothness of $f$,

$$\begin{aligned}
\mathbb{E}\|A\mathbf{x}_n(t)\|^2 &\lesssim \mathbb{E}\|A(\mathbf{x}_n(t) - \mathbf{x}_n^*(t))\|^2 + \mathbb{E}\|A(\mathbf{x}_n^*(t) - \mathbf{x}_*)\|^2 + \|A\mathbf{x}_*\|^2 \\
&\leq L^2\mathbb{E}\|\mathbf{x}_n(t) - \mathbf{x}_n^*(t)\|^2 + \mathbb{E}\|A(\mathbf{x}_n^*(t) - \mathbf{x}_*)\|^2 + L^2\|\mathbf{x}_*\|^2.
\end{aligned}$$ 
(C.4)

Since $U(\cdot)$ is m strongly convex and $\|H\mathbf{x}\|$ is a convex function with respect to $\mathbf{x}$, Theorem 1.1 of Hargé [2004] implies that

$$\mathbb{E}\|A(\mathbf{x}_n^*(t) - \mathbb{E}\mathbf{x}_n^*)\|^2 \leq \mathbb{E}_{\mathbf{x}\sim\mathcal{N}(0,\frac{1}{m})}\|A\mathbf{x}\|^2 = \frac{1}{m}\mathrm{tr}(A^2). \tag{C.5}$$

On the other hand, by Theorem 7 of Basu and DasGupta [1997],

$$(\mathbb{E}\mathbf{x}_n^*(t) - \mathbf{x}_*)^T\Sigma^{-1}(\mathbb{E}\mathbf{x}_n^*(t) - \mathbf{x}_*) \leq 3,$$

where $\Sigma$ is the covariance of $p$ and therefore bounded by $\frac{1}{m}$. Hence

$$\|A(\mathbb{E}\mathbf{x}_n^*(t) - \mathbf{x}_*)\|^2 \leq \frac{\|A\|^2}{m} \leq \frac{1}{m}\mathrm{tr}(A^2). \tag{C.6}$$

Combining equation (C.5) and (C.6), we obtain that

$$\mathbb{E}\|A(\mathbf{x}_n^* - \mathbf{x}_*)\|^2 \lesssim \mathbb{E}\|A(\mathbf{x}_n^* - \mathbb{E}\mathbf{x}_n^*)\|^2 + \|A(\mathbb{E}\mathbf{x}_n^* - \mathbf{x}_*)\|^2 \lesssim \frac{1}{m}\mathrm{tr}(A^2). \tag{C.7}$$

Combining (C.4) and (C.7) and then we achieve the first claim.

To bound $\mathbb{E}\|A(\mathbf{x}_n(t) + \mathbf{v}_n(t))\|^2$, we have

$$\begin{aligned}
\mathbb{E}\|A(\mathbf{x}_n(t) + \mathbf{v}_n(t))\|^2 &\lesssim \mathbb{E}\|A(\mathbf{x}_n(t) + \mathbf{v}_n(t) - \mathbf{x}_n^*(t) - \mathbf{v}_n^*(t))\|^2 \\
&\quad + \mathbb{E}\|A(\mathbf{x}_n^*(t) + \mathbf{v}_n^*(t) - \mathbf{x}_*)\|^2 + \|A\mathbf{x}_*\|^2 \\
&\lesssim L^2\mathbb{E}\|\mathbf{x}_n(t) + \mathbf{v}_n(t) - \mathbf{x}_n^*(t) - \mathbf{v}_n^*(t)\|^2 + \mathbb{E}\|A(\mathbf{x}_n^*(t) - \mathbf{x}_*)\|^2 \\
&\quad + \mathbb{E}\|A\mathbf{v}_n^*(t)\|^2 + L^2\|\mathbf{x}_*\|^2 \\
&\lesssim L^2\mathbb{E}\|\mathbf{x}_n(t) + \mathbf{v}_n(t) - \mathbf{x}_n^*(t) - \mathbf{v}_n^*(t)\|^2 + \frac{1}{m}\mathrm{tr}(A^2) + \frac{1}{m}\mathrm{tr}(A^2) + L^2\|\mathbf{x}_*^2\| \\
&\lesssim L^2\mathbb{E}\|\mathbf{x}_n(t) + \mathbf{v}_n(t) - \mathbf{x}_n^*(t) - \mathbf{v}_n^*(t)\|^2 + \frac{1}{m}\mathrm{tr}(A^2) + L^2\|\mathbf{x}_*\|^2,
\end{aligned}$$

where the second inequality follows by the Lipschitz smoothness of $f(\mathbf{x})$ and Young's inequality; in the third inequality we use (C.7) and $v \sim \mathcal{N}(0, \frac{1}{m})$.

To prove the third claim, we have

$$\begin{aligned}
\mathbb{E}\|A\nabla f(\mathbf{x}_n(t))\|^2 &\lesssim \mathbb{E}\|A(\nabla f(\mathbf{x}_n(t)) - \nabla f(\mathbf{x}_n^*(t)))\|^2 + \mathbb{E}\|A(\nabla f(\mathbf{x}_n^*(t)) - \nabla f(\mathbf{x}_*))\|^2 + \|A\nabla f(\mathbf{x}_*)\|^2 \\
&\lesssim L^4\mathbb{E}\|\mathbf{x}_n(t) - \mathbf{x}_n^*(t)\|^2 + \mathbb{E}L^2\|A(\mathbf{x}_n^*(t) - \mathbf{x}_*)\|^2 + L^2\|\nabla f(\mathbf{x}_*)\|^2 \\
&\lesssim L^4\mathbb{E}\|\mathbf{x}_n(t) - \mathbf{x}_n^*(t)\|^2 + \frac{L^2}{m}\mathrm{tr}(A^2) + L^2 m^2\|\mathbf{x}_*\|^2,
\end{aligned}$$

where in last inequality, we apply (C.7) and $\nabla f(\mathbf{x}_*) + m\mathbf{x}_* = 0$. $\qquad\square$

# D  Technical bounds

In this section, we present some controls that are useful in the proof of Theorem 4.2.

## D.1  Bound of $\mathbb{E}\|\nabla f(\mathbf{x}_n(t)) - \nabla f(\widehat{\mathbf{x}}_n(\alpha))\|^2$

**Lemma D.1.** Let $\mathbf{x}_n(t), \widehat{\mathbf{x}}_n(t)$ be defined as above and $u = \frac{1}{L+m}$. For any fixed $t, \alpha \leq h$.

(i) When Assumption 3.1 hold and $H$ is a uniform upper bound of $\nabla^2 f$, i.e. $\nabla^2 f(\mathbf{x}) \preceq H$ for all $\mathbf{x} \in \mathbb{R}^d$, $\mathbf{x}_n^*(t)$ and $\widehat{\mathbf{x}}_n(\alpha)$ satisfies

$$\mathbb{E}\|\nabla f(\mathbf{x}_n(t)) - \nabla f(\widehat{\mathbf{x}}_n(\alpha))\|^2 \lesssim h^2 L^2 \mathbb{E}\Omega_n + \frac{h^2}{m}\mathrm{tr}(H^2) + L^2 h^2 \|\mathbf{x}_*\|^2.$$

(ii) When Assumptions 3.1, E.1 and E.2 hold, we have

$$\mathbb{E}\|\nabla f(\mathbf{x}_n(t)) - \nabla f(\widehat{\mathbf{x}}_n(\alpha))\|^2 \lesssim h^2 L^2 \mathbb{E}\Omega_n + \frac{h^2 L}{m}M + L^2 h^2 \|\mathbf{x}_*\|^2 + L_2^2 u^2 h^6 d^2.$$

*Proof.* For any $\mathbf{x}, \mathbf{y} \in \mathbb{R}^d$, we have

$$\|\nabla f(\mathbf{x}) - \nabla f(\mathbf{y})\|^2 = \left\|\int_0^1 \nabla^2 f((1-k)\mathbf{x} + k\mathbf{y})(\mathbf{x} - \mathbf{y})\mathrm{d}k\right\|^2. \tag{D.1}$$

Denote $\bar{\mathcal{H}} = \int_0^1 \nabla^2 f(\mathbf{x} + k(\mathbf{y} - \mathbf{x}))\mathrm{d}k$ and $\bar{\mathcal{H}}$ depends on $\mathbf{x}$ and $\mathbf{y}$. Substitute $\mathbf{x}, \mathbf{y}$ by $\mathbf{x}_n(t), \widehat{\mathbf{x}}_n(\alpha)$, and then we obtain $\|\nabla f(\mathbf{x}_n(t)) - \nabla f(\widehat{\mathbf{x}}_n(\alpha))\|^2 = \|\bar{\mathcal{H}}(\mathbf{x}_n(t) - \widehat{\mathbf{x}}_n(\alpha))\|^2$. Then by plugging in the close-form solution into $\|\bar{\mathcal{H}}(\mathbf{x}_n(t) - \widehat{\mathbf{x}}_n(\alpha))\|^2$,

$$
\begin{aligned}
&\|\bar{\mathcal{H}}(\mathbf{x}_n(t) - \widehat{\mathbf{x}}_n(\alpha))\|^2\\
&\overset{a}{=}\left\|\bar{\mathcal{H}}\Big((A_{11}(t) - A_{11}(\alpha) - A_{12}(t) + A_{12}(\alpha))\mathbf{x}_n + (A_{12}(t) - A_{12}(\alpha))(\mathbf{x}_n + \mathbf{v}_n)+\right.\\
&\quad u\int_0^t A_{12}(s-t)\nabla f(\widehat{\mathbf{x}}_n(\alpha))\mathrm{d}s - u\int_0^\alpha A_{12}(s-\alpha)\nabla f(\mathbf{x}_n)\mathrm{d}s + 2\sqrt{u}\int_0^t A_{12}(s-t)\mathrm{d}\mathbf{B}_s\\
&\quad\left.- 2\sqrt{u}\int_0^\alpha A_{12}(s-\alpha)\mathrm{d}\mathbf{B}_s\Big)\right\|^2\\
&\overset{b}{\lesssim}(A_{11}(t) - A_{11}(\alpha) - A_{12}(t) + A_{12}(\alpha))^2\|\bar{\mathcal{H}}\mathbf{x}_n\|^2 + (A_{12}(t) - A_{12}(\alpha))^2\|\bar{\mathcal{H}}(\mathbf{x}_n + \mathbf{v}_n)\|^2\\
&\quad + u^2\underbrace{\left\|\int_0^t A_{12}(s-t)\bar{\mathcal{H}}\nabla f(\widehat{\mathbf{x}}_n(\alpha))\mathrm{d}s - \int_0^\alpha A_{12}(s-\alpha)\bar{\mathcal{H}}\nabla f(\mathbf{x}_n)\mathrm{d}s\right\|^2}_{①}\\
&\quad + u\underbrace{\left\|\int_0^t A_{12}(s-t)\bar{\mathcal{H}}\mathrm{d}\mathbf{B}_s - \int_0^\alpha A_{12}(s-\alpha)\bar{\mathcal{H}}\mathrm{d}\mathbf{B}_s\right\|^2}_{②},
\end{aligned}
\tag{D.2}
$$

where in $\overset{a}{=}$ we use Lemma 4.1; $\overset{b}{\lesssim}$ follows by the Jensen's inequality. Besides Lemma A.1 implies that $(A_{11}(t) - A_{11}(\alpha) - A_{12}(t) + A_{12}(\alpha))^2\|H\mathbf{x}_n\|^2 \lesssim h^2\|H\mathbf{x}_n\|^2$ and $(A_{12}(t) - A_{12}(\alpha))^2\|H(\mathbf{x}_n + \mathbf{v}_n)\|^2 \lesssim h^2\|H(\mathbf{x}_n + \mathbf{v}_n)\|^2$.

For the expectation of term ①, we have

$$
\mathbb{E}\left\|\int_0^t A_{12}(s-t)\bar{\mathcal{H}}\nabla f(\widehat{\mathbf{x}}_n(\alpha))\mathrm{d}s - \int_0^\alpha A_{12}(s-\alpha)\bar{\mathcal{H}}\nabla f(\mathbf{x}_n)\mathrm{d}s\right\|^2
$$

$$
\lesssim \mathbb{E}\left\|\int_0^t A_{12}(s-t)\bar{\mathcal{H}}\big(\nabla f(\widehat{\mathbf{x}}_n(\alpha)) - \nabla f(\mathbf{x}_n)\big)\mathrm{d}s\right\|^2 +
$$

$$
\mathbb{E}\left\|\int_\alpha^t A_{12}(s-\alpha)\bar{\mathcal{H}}\nabla f(\mathbf{x}_n)\mathrm{d}s\right\|^2 + \mathbb{E}\left\|\int_0^t (A_{12}(s-t) - A_{12}(s-\alpha))\bar{\mathcal{H}}\nabla f(\mathbf{x}_n)\mathrm{d}s\right\|^2
$$

$$
\leq \mathbb{E}t\int_0^t A_{12}(s-t)^2\mathrm{d}s\|\bar{\mathcal{H}}\big(\nabla f(\widehat{\mathbf{x}}_n(\alpha)) - \nabla f(\mathbf{x}_n)\big)\|^2
$$

$$
+ \mathbb{E}\left|(\alpha-t)\int_\alpha^t A_{12}(s-\alpha)^2\mathrm{d}s\right|\|\bar{\mathcal{H}}\nabla f(\mathbf{x}_n)\|^2
$$

$$
+ \mathbb{E}t\int_0^t (A_{12}(s-t) - A_{12}(s-\alpha))^2\mathrm{d}s\|\bar{\mathcal{H}}\nabla f(\mathbf{x}_n)\|^2
$$

$$
\lesssim h^4\mathbb{E}\|\bar{\mathcal{H}}\big(\nabla f(\widehat{\mathbf{x}}_n(\alpha)) - \nabla f(\mathbf{x}_n)\big)\|^2 + h^4\mathbb{E}\|\bar{\mathcal{H}}\nabla f(\mathbf{x}_n)\|^2,
\tag{D.3}
$$

where in the last inequality we use that $t, \alpha < h$ and $A_{12}(t) = \mathcal{O}(t)$. Let $\bar{\mathcal{H}}' = \int_0^1 \nabla^2 f(\widehat{\mathbf{x}}_n(\alpha) + k(\widehat{\mathbf{x}}_n(\alpha) - \mathbf{x}_n))\mathrm{d}k$. By the definition of $\widehat{\mathbf{x}}_n(\alpha)$, $\mathbb{E}\|\bar{\mathcal{H}}\big(\nabla f(\widehat{\mathbf{x}}_n(\alpha)) - \nabla f(\mathbf{x}_n)\big)\|^2$ can be upper bounded by

$$
\mathbb{E}\|\bar{\mathcal{H}}\big(\nabla f(\widehat{\mathbf{x}}_n(\alpha)) - \nabla f(\mathbf{x}_n)\big)\|^2
$$

$$
\leq L^2\mathbb{E}\|\big(\nabla f(\widehat{\mathbf{x}}_n(\alpha)) - \nabla f(\mathbf{x}_n)\big)\|^2
$$

$$
= L^2\mathbb{E}\left\|\bar{\mathcal{H}}'\big((A_{11}(\alpha) - 1)\mathbf{x}_n + A_{12}(\alpha)\mathbf{v}_n + u\int_0^\alpha A_{12}(t-\alpha)\nabla f(\mathbf{x}_n)\mathrm{d}t + 2\sqrt{u}\int_0^\alpha A_{12}(t-\alpha)\mathrm{d}\mathbf{B}_t\big)\right\|^2
$$

$$
\lesssim L^2\mathbb{E}A_{12}(\alpha)^2\|\bar{\mathcal{H}}'(\mathbf{x}_n + \mathbf{v}_n)\|^2 + \mathbb{E}(A_{11}(x) - A_{12}(x) - 1)^2L^2\|\bar{\mathcal{H}}'\mathbf{x}_n\|^2
$$

$$
+ u^2\mathbb{E}\alpha\int_0^\alpha L^2\|A_{12}(t-\alpha)\bar{\mathcal{H}}'\nabla f(\mathbf{x}_n)\|^2\mathrm{d}t + u\mathbb{E}L^2\left\|\int_0^\alpha A_{12}(t-\alpha)\bar{\mathcal{H}}'\mathrm{d}\mathbf{B}_t\right\|^2
$$

$$
\lesssim h^2L^2\mathbb{E}\|\bar{\mathcal{H}}'(\mathbf{x}_n + \mathbf{v}_n)\|^2 + h^2L^2\mathbb{E}\|\bar{\mathcal{H}}'\mathbf{x}_n\|^2 + u^2h^4L^2\mathbb{E}\|\bar{\mathcal{H}}'\nabla f(\mathbf{x}_n)\|^2
$$

$$
+ uL^2\mathbb{E}\left\|\int_0^\alpha A_{12}(t-\alpha)\bar{\mathcal{H}}'\mathrm{d}\mathbf{B}_t\right\|^2,
$$

where in the last inequality we apply Lemma A.1. Note that $\mathbf{x}_n(0) = \mathbf{x}_n$ and $\mathbf{v}_n(0) = \mathbf{v}_n$. Then by Lemma C.2,

$$
\mathbb{E}\|\bar{\mathcal{H}}(\nabla f(\widehat{\mathbf{x}}_n(\alpha)) - \nabla f(\mathbf{x}_n))\|^2
$$

$$
\leq L^2\mathbb{E}\|\nabla f(\widehat{\mathbf{x}}_n(\alpha)) - \nabla f(\mathbf{x}_n)\|^2
$$

$$
\lesssim h^2(L^4\mathbb{E}\|\mathbf{x}_n + \mathbf{v}_n - \mathbf{x}_n^* - \mathbf{v}_n^*\|^2 + \frac{L^2}{m}\mathbb{E}\mathrm{tr}(\bar{\mathcal{H}}'^2) + L^4\|\mathbf{x}_*\|^2) + h^2(L^4\mathbb{E}\|\mathbf{x}_n - \mathbf{x}_n^*\|^2
$$

$$
+ \frac{L^2}{m}\mathbb{E}\mathrm{tr}(\bar{\mathcal{H}}'^2) + L^2\|\mathbf{x}_*\|^2) + u^2h^2(L^4\mathbb{E}\|\mathbf{x}_n - \mathbf{x}_n^*\|^2 + \frac{L^2}{m}\mathbb{E}\mathrm{tr}(\bar{\mathcal{H}}'^2) + L^4m^2\|\mathbf{x}_*\|^2) \quad \text{(D.4)}
$$

$$
+ uL^2\mathbb{E}\left\|\int_0^\alpha A_{12}(t-\alpha)\bar{\mathcal{H}}'\mathrm{d}\mathbf{B}_t\right\|^2
$$

$$
\lesssim h^2L^4\mathbb{E}\Omega_n + \frac{h^2L^2}{m}\mathbb{E}\mathrm{tr}(\bar{\mathcal{H}}'^2) + h^2L^4\|\mathbf{x}_*\|^2 + uL^2\mathbb{E}\left\|\int_0^\alpha A_{12}(t-\alpha)\bar{\mathcal{H}}'\mathrm{d}\mathbf{B}_t\right\|^2,
$$

where in the last inequality we use $u = \frac{1}{L+m} \leq \min\left\{\frac{1}{L}, \frac{1}{m}\right\}$. Plugging (D.4) into (D.3) and then we have an upper bound of the expectation of term ①. Plugging the upper bound of term ① into (D.2),

and thus we have the following inequality

$$
\mathbb{E}\|\nabla f(\mathbf{x}_n(t)) - \nabla f(\widehat{\mathbf{x}}_n(\alpha))\|^2
$$
$$
\lesssim h^2 \mathbb{E}\|\bar{\mathcal{H}}\mathbf{x}_n\|^2 + h^2 \mathbb{E}\|\bar{\mathcal{H}}(\mathbf{x}_n + \mathbf{v}_n)\|^2 + u^2 h^4 \mathbb{E}\|\bar{\mathcal{H}}\nabla f(\mathbf{x}_n)\|^2
$$
$$
+ u^2 h^4 \left( h^2 L^4 \mathbb{E}\Omega_n + \frac{h^2 L^2}{m}\mathbb{E}\mathrm{tr}\left((\bar{\mathcal{H}}')^2\right) + h^2 L^4 \|\mathbf{x}_*\|^2 + u\mathbb{E}L^2 \left\|\int_0^\alpha A_{12}(t-\alpha)\bar{\mathcal{H}}'\mathrm{d}\mathbf{B}_t\right\|^2 \right)
$$
$$
+ u\mathbb{E}\left\|\int_0^t A_{12}(s-t)\bar{\mathcal{H}}\mathrm{d}\mathbf{B}_s - \int_0^\alpha A_{12}(s-\alpha)\bar{\mathcal{H}}\mathrm{d}\mathbf{B}_s\right\|^2
$$
$$
\lesssim h^2 L^2 \mathbb{E}\Omega_n + \frac{h^2}{m}\mathbb{E}\left(\mathrm{tr}\left((\bar{\mathcal{H}}')^2\right) + \mathrm{tr}\left(\bar{\mathcal{H}}^2\right)\right) + L^2 h^2 \|\mathbf{x}_*\|^2 + u^3 h^4 L^2 \mathbb{E}\left\|\int_0^\alpha A_{12}(t-\alpha)\bar{\mathcal{H}}'\mathrm{d}\mathbf{B}_t\right\|^2
$$
$$
+ u\mathbb{E}\left\|\int_0^t A_{12}(s-t)\bar{\mathcal{H}}\mathrm{d}\mathbf{B}_s - \int_0^\alpha A_{12}(s-\alpha)\bar{\mathcal{H}}\mathrm{d}\mathbf{B}_s\right\|^2,
$$

where in the last inequality we use Lemma C.2 and the definition of $\Omega_n$.

When we assume $\nabla^2 f(\mathbf{x})$ has a uniform upper bound $H$, we have

$$
u\mathbb{E}\left\|\int_0^t A_{12}(s-t)\bar{\mathcal{H}}\mathrm{d}\mathbf{B}_s - \int_0^\alpha A_{12}(s-\alpha)\bar{\mathcal{H}}\mathrm{d}\mathbf{B}_s\right\|^2
$$
$$
\lesssim u\mathbb{E}\left\|\int_0^h hH\mathrm{d}\mathbf{B}_s\right\|^2 \lesssim \frac{h^3}{m}\mathrm{tr}(H^2) \leq \frac{h^3 L}{m}\mathrm{tr}(H) \tag{D.5}
$$

and

$$
u^3 h^4 L^2 \mathbb{E}\left\|\int_0^\alpha A_{12}(t-\alpha)\bar{\mathcal{H}}'\mathrm{d}\mathbf{B}_t\right\|^2 \leq u^3 h^4 L^2 h^3 \mathrm{tr}(H^2) \lesssim \frac{h^4}{m}\mathrm{tr}(H^2) \leq \frac{h^4 L}{m}\mathrm{tr}(H).
$$

Besides we have $\mathrm{tr}\left((\bar{\mathcal{H}}')^2\right) + \mathrm{tr}\left(\bar{\mathcal{H}}^2\right) \leq L\mathrm{tr}\left(\bar{\mathcal{H}}'\right) + L\mathrm{tr}\left(\bar{\mathcal{H}}\right) \lesssim L\mathrm{tr}(H)$. These bounds yield the control

$$
\mathbb{E}\|\nabla f(\mathbf{x}_n(t)) - \nabla f(\widehat{\mathbf{x}}_n(\alpha))\|^2 \lesssim h^2 L^2 \mathbb{E}\Omega_n + \frac{h^2 L}{m}\mathrm{tr}(H) + L^2 h^2 \|\mathbf{x}_*\|^2. \tag{D.6}
$$

And when $f$ is $L_2$-Hessian smooth, by Lemmas D.2 and D.3, we have

$$
\mathbb{E}\|\nabla f(\mathbf{x}_n(t)) - \nabla f(\widehat{\mathbf{x}}_n(\alpha))\|^2
$$
$$
\lesssim h^2 L^2 \mathbb{E}\Omega_n + \frac{h^2}{m}\mathbb{E}\left(\mathrm{tr}\left((\bar{\mathcal{H}}')^2\right) + \mathrm{tr}\left(\bar{\mathcal{H}}^2\right)\right) + uh^3 LM + L^2 h^2 \|\mathbf{x}_*\|^2
$$
$$
+ L_2^2 u^2 h^6 d^2 + L_2^2 u^4 L^2 h^{10} d^2
$$
$$
\lesssim h^2 L^2 \mathbb{E}\Omega_n + L\frac{h^2}{m}\mathbb{E}\left(\mathrm{tr}\left(\bar{\mathcal{H}}'\right) + \mathrm{tr}\left(\bar{\mathcal{H}}\right)\right) + \frac{h^2 L}{m}M + L^2 h^2 \|\mathbf{x}_*\|^2 + L_2^2 u^2 h^6 d^2 \tag{D.7}
$$
$$
\lesssim h^2 L^2 \mathbb{E}\Omega_n + \frac{Lh^2 M}{m} + L^2 h^2 \|\mathbf{x}_*\|^2 + L_2^2 u^2 h^6 d^2.
$$

$\qquad\square$

**Lemma D.2.** Let $\bar{\mathcal{H}}$ be defined as above. When Assumptions E.1 and E.2 hold, we have

$$
\mathbb{E}\left\|\int_0^t A_{12}(s-t)\bar{\mathcal{H}}\mathrm{d}\mathbf{B}_s - \int_0^\alpha A_{12}(s-\alpha)\bar{\mathcal{H}}\mathrm{d}\mathbf{B}_s\right)\right\|^2 \lesssim h^3 LM + L_2^2 uh^6 d^2.
$$

*Proof.* First, we have

$$\mathbb{E}\left\|\left(\int_0^t A_{12}(s-t)\bar{\mathcal{H}}\mathrm{d}\mathbf{B}_s - \int_0^\alpha A_{12}(s-\alpha)\bar{\mathcal{H}}\mathrm{d}\mathbf{B}_s\right)\right\|^2$$

$$=\mathbb{E}\left\|\bar{\mathcal{H}}\left(\int_0^t A_{12}(s-t)\mathrm{d}\mathbf{B}_s - \int_0^\alpha A_{12}(s-\alpha)\mathrm{d}\mathbf{B}_s\right)\right\|^2$$

$$\overset{\mathrm{a}}{\lesssim}\mathbb{E}\left\|\left(\bar{\mathcal{H}} - \mathbb{E}\left[\bar{\mathcal{H}}|\mathcal{F}_n\right]\right)\left(\int_0^t A_{12}(s-t)\mathrm{d}\mathbf{B}_s - \int_0^\alpha A_{12}(s-\alpha)\mathrm{d}\mathbf{B}_s\right)\right\|^2$$

$$+\mathbb{E}\left\|\mathbb{E}\left[\bar{\mathcal{H}}|\mathcal{F}_n\right]\left(\int_0^t A_{12}(s-t)\mathrm{d}\mathbf{B}_s - \int_0^\alpha A_{12}(s-\alpha)\mathrm{d}\mathbf{B}_s\right)\right\|^2 \qquad (\mathrm{D}.8)$$

$$\lesssim\mathbb{E}\left(\left\|\bar{\mathcal{H}} - \mathbb{E}\left[\bar{\mathcal{H}}|\mathcal{F}_n\right]\right\|^2\left\|\left(\int_0^t A_{12}(s-t)\mathrm{d}\mathbf{B}_s - \int_0^\alpha A_{12}(s-\alpha)\mathrm{d}\mathbf{B}_s\right)\right\|^2\right)$$

$$+\mathbb{E}\left\|\mathbb{E}\left[\bar{\mathcal{H}}|\mathcal{F}_n\right]\left(\int_0^t A_{12}(s-t)\mathrm{d}\mathbf{B}_s - \int_0^\alpha A_{12}(s-\alpha)\mathrm{d}\mathbf{B}_s\right)\right\|^2,$$

where $\overset{\mathrm{a}}{\lesssim}$ follows by dividing $\bar{\mathcal{H}}$ into $\mathbb{E}\left[\bar{\mathcal{H}}|\mathcal{F}_n\right]$ and $\bar{\mathcal{H}} - \mathbb{E}\left[\bar{\mathcal{H}}|\mathcal{F}_n\right]$. In the last inequality, $\mathbb{E}\left[\bar{\mathcal{H}}|\mathcal{F}_n\right]$ is independent of $\mathbf{B}_s$, and thus

$$\mathbb{E}\left\|\mathbb{E}\left[\bar{\mathcal{H}}|\mathcal{F}_n\right]\left(\int_0^t A_{12}(s-t)\mathrm{d}\mathbf{B}_s - \int_0^\alpha A_{12}(s-\alpha)\mathrm{d}\mathbf{B}_s'\right)\right\|^2$$

$$\lesssim h^2\mathbb{E}\left\|\mathbb{E}\left[\bar{\mathcal{H}}|\mathcal{F}_n\right]\int_0^h \mathrm{d}\mathbf{B}_s\right\|^2 \qquad (\mathrm{D}.9)$$

$$\leq h^3\mathbb{E}\mathrm{tr}\left(\left(\mathbb{E}\left[\bar{\mathcal{H}}|\mathcal{F}_n\right]\right)^2\right).$$

As for the firs term, let $\{\mathbf{B}_s'\}_{0\leq s\leq h}$ be a standard Brownian motion which is independent of $\{\mathbf{B}_s\}_{0\leq s\leq h}$. Define $\widehat{\mathbf{x}}_n'(\alpha) = \mathcal{J}(\alpha, \mathbf{x}_n; \{\mathbf{B}_s'\}_{0\leq s\leq\alpha}, (\mathbf{x}_n, \mathbf{v}_n))$ and $\mathbf{x}_n'(t) = \mathcal{J}(\alpha, \widehat{\mathbf{x}}_n'(\alpha); \{\mathbf{B}_s'\}_{0\leq s\leq t}, (\mathbf{x}_n, \mathbf{v}_n))$. Then we have

$$\mathbb{E}\left\|\bar{\mathcal{H}} - \mathbb{E}\left[\bar{\mathcal{H}}|\mathcal{F}_n\right]\right\|^2 \overset{\mathrm{a}}{\leq}\mathbb{E}\int_0^1 \left\|\nabla^2 f((1-k)\mathbf{x}_n(t) + k\widehat{\mathbf{x}}_n(\alpha)) - \nabla^2 f((1-k)\mathbf{x}_n'(t) + k\widehat{\mathbf{x}}_n'(\alpha))\right\|^2 \mathrm{d}k$$

$$\overset{\mathrm{b}}{\leq}L_2^2\mathbb{E}\int_0^1 \|(1-k)(\mathbf{x}_n(t) - \mathbf{x}_n'(t)) + k(\widehat{\mathbf{x}}_n(\alpha) - \widehat{\mathbf{x}}_n'(\alpha))\|^2\mathrm{d}k$$

$$\overset{\mathrm{c}}{\lesssim}L_2^2 u^2\mathbb{E}\left\|\int_0^t A_{12}(s-t)\left(\nabla f(\widehat{\mathbf{x}}_n(\alpha)) - \nabla f(\widehat{\mathbf{x}}_n'(\alpha))\right)\mathrm{d}s\right\|^2$$

$$+L_2^2 u\mathbb{E}\left\|\int_0^\alpha A_{12}(s-\alpha)\mathrm{d}\mathbf{B}_s - \int_0^\alpha A_{12}(s-\alpha)\mathrm{d}\mathbf{B}_s'\right\|^2$$

$$+L_2^2 u\mathbb{E}\left\|\int_0^t A_{12}(s-t)\mathrm{d}\mathbf{B}_s - \int_0^t A_{12}(s-t)\mathrm{d}\mathbf{B}_s'\right\|^2, \qquad (\mathrm{D}.10)$$

where $\overset{\mathrm{a}}{\leq}$ follows by the definition of $\bar{\mathcal{H}}$ and Jensen's inequality; in $\overset{\mathrm{b}}{\leq}$ we use the Hessian smoothness of $f$; $\overset{\mathrm{c}}{\lesssim}$ follows by the definition of $\mathbf{x}_n(t), \mathbf{x}_n'(t), \widehat{\mathbf{x}}_n(\alpha)$ and $\widehat{\mathbf{x}}_n'(\alpha)$. For the first term of the last inequality, by the definition of $\widehat{\mathbf{x}}_n(\alpha)$ and $\widehat{\mathbf{x}}_n'(\alpha)$, we have

$$\left\|\int_0^t A_{12}(s-t)\left(\nabla f(\widehat{\mathbf{x}}_n(\alpha)) - \nabla f(\widehat{\mathbf{x}}_n'(\alpha))\right)\mathrm{d}s\right\|^2$$

$$\lesssim L^2 h^4 u\left\|\int_0^\alpha A_{12}(s-\alpha)\mathrm{d}\mathbf{B}_s - \int_0^\alpha A_{12}(s-\alpha)\mathrm{d}\mathbf{B}_s'\right\|^2.$$

And thus we have

$$\mathbb{E}\left(\left\|\bar{\mathcal{H}} - \mathbb{E}\left[\bar{\mathcal{H}}|\mathcal{F}_n\right]\right\|^2 \left\|\left(\int_0^t A_{12}(s-t)\mathrm{d}\mathbf{B}_s - \int_0^\alpha A_{12}(s-\alpha)\mathrm{d}\mathbf{B}_s\right)\right\|^2\right)$$

$$\lesssim L_2^2 u \mathbb{E}\left\|\int_0^t A_{12}(s-t)\mathrm{d}\mathbf{B}_s\right\|^4 + \mathbb{E}L_2^2 u\left\|\int_0^t A_{12}(s-t)\mathrm{d}\mathbf{B}'_s\right\|^4 \tag{D.11}$$

$$+ L_2^2 u \mathbb{E}\left\|\int_0^\alpha A_{12}(s-\alpha)\mathrm{d}\mathbf{B}_s\right\|^4 + L_2^2 u \mathbb{E}\left\|\int_0^\alpha A_{12}(s-\alpha)\mathrm{d}\mathbf{B}'_s\right\|^4$$

$$\lesssim L_2^2 u h^6 d^2.$$

Plugging (D.11) and (D.9) in to control (D.8) yields

$$\mathbb{E}\left\|\int_0^t A_{12}(s-t)\bar{\mathcal{H}}\mathrm{d}\mathbf{B}_s - \int_0^\alpha A_{12}(s-\alpha)\bar{\mathcal{H}}\mathrm{d}\mathbf{B}_s\right)\right\|^2$$

$$\lesssim h^3 \mathbb{E}\mathrm{tr}\left(\left(\mathbb{E}[\bar{\mathcal{H}}|\mathcal{F}_n]\right)^2\right) + L_2^2 u h^6 d^2$$

$$\leq h^3 LM + L_2^2 u h^6 d^2,$$

where the last inequality follows by Assumption E.2 and the observation that $\mathbb{E}\mathrm{tr}\left(\left(\mathbb{E}[\bar{\mathcal{H}}|\mathcal{F}_n]\right)^2\right) \leq L\mathbb{E}\mathbb{E}[\mathrm{tr}\left(\bar{\mathcal{H}}\right)|\mathcal{F}_n] \leq LM$. $\qquad\square$

**Lemma D.3.** Let $\bar{\mathcal{H}}$ and $\bar{\mathcal{H}}'$ be defined as above. When Assumptions E.2 and E.1 hold, we have

$$uL^2\mathbb{E}\left\|\int_0^\alpha A_{12}(t-\alpha)\bar{\mathcal{H}}'\mathrm{d}\mathbf{B}_t\right\|^2 \lesssim uL^3 h^3 M + L_2^2 u^2 L^2 h^6 d^2.$$

*Proof.* Similar to the proof of Lemma D.2, we have

$$uL^2\mathbb{E}\left\|\int_0^\alpha A_{12}(t-\alpha)\bar{\mathcal{H}}'\mathrm{d}\mathbf{B}_t\right\|^2$$

$$= uL^2\mathbb{E}\left\|\bar{\mathcal{H}}'\int_0^\alpha A_{12}(t-\alpha)\mathrm{d}\mathbf{B}_t\right\|^2$$

$$\lesssim uL^2\mathbb{E}\left\|\left(\bar{\mathcal{H}}' - \mathbb{E}\left[\bar{\mathcal{H}}'|\mathcal{F}_n\right]\right)\int_0^\alpha A_{12}(t-\alpha)\mathrm{d}\mathbf{B}_t\right\|^2 + uL^2\mathbb{E}\left\|\mathbb{E}\left[\bar{\mathcal{H}}'|\mathcal{F}_n\right]\int_0^\alpha A_{12}(t-\alpha)\mathrm{d}\mathbf{B}_t\right\|^2$$

$$\lesssim uL^2\mathbb{E}\left(\left\|\bar{\mathcal{H}}' - \mathbb{E}\left[\bar{\mathcal{H}}'|\mathcal{F}_n\right]\right\|^2 \left\|\int_0^\alpha A_{12}(t-\alpha)\mathrm{d}\mathbf{B}_t\right\|^2\right) + uL^2 h^3 \mathbb{E}\mathrm{tr}\left(\left(\mathbb{E}\left[\bar{\mathcal{H}}'|\mathcal{F}_n\right]\right)^2\right).$$

$$\tag{D.12}$$

We follow the notation of $\mathbf{B}'_s$ and $\widehat{\mathbf{x}}'_n(\alpha)$ in (D.10). Then

$$\mathbb{E}\left(\left\|\bar{\mathcal{H}}' - \mathbb{E}\left[\bar{\mathcal{H}}'|\mathcal{F}_n\right]\right\|^2 \left\|\int_0^\alpha A_{12}(t-\alpha)\mathrm{d}\mathbf{B}_t\right\|^2\right)$$

$$\lesssim \mathbb{E}\left(L_2^2 \int_0^1 (1-k)^2 \|\widehat{\mathbf{x}}_n(\alpha) - \widehat{\mathbf{x}}'_n(\alpha)\|^2 \mathrm{d}k \left\|\int_0^\alpha A_{12}(t-\alpha)\mathrm{d}\mathbf{B}_t\right\|^2\right)$$

$$\lesssim L_2^2 \mathbb{E}\left(u\left\|\int_0^\alpha A_{12}(s-\alpha)\mathrm{d}\mathbf{B}_s - \int_0^\alpha A_{12}(s-\alpha)\mathrm{d}\mathbf{B}'_s\right\|^2 \left\|\int_0^\alpha A_{12}(t-\alpha)\mathrm{d}\mathbf{B}_t\right\|^2\right) \tag{D.13}$$

$$\lesssim L_2^2 u \mathbb{E}\left\|\int_0^\alpha A_{12}(s-\alpha)\mathrm{d}\mathbf{B}_s\right\|^4$$

$$\lesssim L_2^2 u h^6 d^2.$$

Combining (D.12) and (D.13) and using $\mathrm{tr}\left(\mathbb{E}[\bar{\mathcal{H}}'|\mathcal{F}_n]\right) \le LM$ yields

$$uL^2\mathbb{E}\left\|\int_0^\alpha A_{12}(t-\alpha)\bar{\mathcal{H}}'\mathrm{d}\mathbf{B}_t\right\|^2 \lesssim uL^3h^3M + L_2^2u^2L^2h^6d^2.$$

$\square$

## D.2  Bound of $\mathbb{E}\|\mathbf{x}_n(t) - \mathbf{x}_n - \mathbf{x}_n^*(t) + \mathbf{x}_n^* + \mathbf{v}_n(t) - \mathbf{v}_n - \mathbf{v}_n^*(t) + \mathbf{v}_n^*\|^2$

To prove Lemma 5.3, we consider the following Lemma which also includes the Hessian smooth case.

**Lemma D.4.** Let $\mathbf{x}_n(t), \mathbf{v}_n(t), \mathbf{v}_n^*(t)$ and $\mathbf{v}_n^*(t)$ be defined as above and $u = \frac{1}{L+m}$.

(i) Under Assumption 3.1, for any $t, \alpha \le h$, we have

$$\mathbb{E}\|\mathbf{x}_n(t) - \mathbf{x}_n - \mathbf{x}_n^*(t) + \mathbf{x}_n^* + \mathbf{v}_n(t) - \mathbf{v}_n - \mathbf{v}_n^*(t) + \mathbf{v}_n^*\|^2$$
$$\lesssim h^2\mathbb{E}\mathbb{E}_{t\sim\rho}\Omega_n(t) + h^4\mathbb{E}\Omega_n + u^2h^4\frac{L}{m}\mathrm{tr}(H) + h^4\|\mathbf{x}_*\|^2. \tag{D.14}$$

(ii) Under Assumptions 3.1, E.1 and E.2, for any $t, \alpha \le h$, we have

$$\mathbb{E}\|\mathbf{x}_n(t) - \mathbf{x}_n - \mathbf{x}_n^*(t) + \mathbf{x}_n^* + \mathbf{v}_n(t) - \mathbf{v}_n - \mathbf{v}_n^*(t) + \mathbf{v}_n^*\|^2$$
$$\lesssim h^2\mathbb{E}\mathbb{E}_{t\sim\rho}\Omega_n(t) + h^4\mathbb{E}\Omega_n + u^2\frac{h^4LM}{m} + h^4\|\mathbf{x}_*\|^2 + L_2^2u^4h^8d^2.$$

*Proof.* Since the discretized process and the continuous process are synchronously coupled,

$$\mathbf{x}_n(t) - \mathbf{x}_n - \mathbf{x}_n^*(t) + \mathbf{x}_n^* + \mathbf{v}_n(t) - \mathbf{v}_n - \mathbf{v}_n^*(t) + \mathbf{v}_n^*$$
$$= \int_0^t (\mathbf{v}_n(s) - \mathbf{v}_n^*(s) - um(\mathbf{x}_n(s) - \mathbf{x}_n^*(s)) - 2(\mathbf{v}_n(s) - \mathbf{v}_n^*(s)) - u\nabla f(\widehat{\mathbf{x}}_n(\alpha)) + u\nabla f(\mathbf{x}_n^*(s)))\mathrm{d}s. \tag{D.15}$$

Hence the square norm satisfies

$$\|\mathbf{x}_n(t) - \mathbf{x}_n - \mathbf{x}_n^*(t) + \mathbf{x}_n^* + \mathbf{v}_n(t) - \mathbf{v}_n - \mathbf{v}_n^*(t) + \mathbf{v}_n^*\|^2$$
$$\overset{\mathrm{a}}{\le} \left(3t\int_0^t \|\mathbf{v}_n(s) - \mathbf{v}_n^*(s) + \mathbf{x}_n(s) - \mathbf{x}_n^*(s)\|^2\mathrm{d}s + 3(1-um)^2t\int_0^t \|\mathbf{x}_n(s) - \mathbf{x}_n^*(s)\|^2\mathrm{d}s \right.$$
$$\left. + 3u^2t\int_0^t \|\nabla f(\widehat{\mathbf{x}}_n(\alpha)) - \nabla f(\mathbf{x}_n^*(s))\|^2\mathrm{d}s\right)$$
$$\overset{\mathrm{b}}{\le} 3t\int_0^t \Omega_n(s)\mathrm{d}s + 3u^2t\int_0^t \|\nabla f(\widehat{\mathbf{x}}_n(\alpha)) - \nabla f(\mathbf{x}_n^*(s))\|^2\mathrm{d}s, \tag{D.16}$$

where $\overset{\mathrm{a}}{\le}$ follows by equation [D.15] and Jensen's inequality; in $\overset{\mathrm{b}}{\le}$ we use $|1 - um| \le 1$ and substitute the square norm by $\Omega_n(s)$. To bound the second term in the last inequality, we have

$$\int_0^t \|\nabla f(\widehat{\mathbf{x}}_n(\alpha)) - \nabla f(\mathbf{x}_n^*(s))\|^2\mathrm{d}s$$
$$\lesssim \int_0^t \|\nabla f(\mathbf{x}_n(s)) - \nabla f(\mathbf{x}_n^*(s))\|^2\mathrm{d}s + \int_0^t \|\nabla f(\widehat{\mathbf{x}}_n(\alpha)) - \nabla f(\mathbf{x}_n(s))\|^2\mathrm{d}s \tag{D.17}$$
$$\le L^2\int_0^t \Omega_n(s)\mathrm{d}s + \int_0^t \|\nabla f(\widehat{\mathbf{x}}_n(\alpha)) - \nabla f(\mathbf{x}_n(s))\|^2\mathrm{d}s,$$

where in the second inequality, we use the Lipschitz smoothness of $f$ and $\|\mathbf{x}_n(t) - \mathbf{x}_n^*(t)\| \leq \Omega_n(t)$. By combining bound (D.16) and (D.17), we arrive at that

$$\mathbb{E}\|\mathbf{x}_n(t) - \mathbf{x}_n - \mathbf{x}_n^*(t) + \mathbf{x}_n^* + \mathbf{v}_n(t) - \mathbf{v}_n - \mathbf{v}_n^*(t) + \mathbf{v}_n^*\|^2$$

$$\lesssim t \int_0^t \mathbb{E}\Omega_n(s)\mathrm{d}s + u^2 t \int_0^t \mathbb{E}\|\nabla f(\widehat{\mathbf{x}}_n(\alpha)) - \nabla f(\mathbf{x}_n(s))\|^2 \mathrm{d}s \qquad \text{(D.18)}$$

$$\leq h^2 \int_0^h \frac{1}{h}\mathbb{E}\Omega_n(t)\mathrm{d}t + u^2 h \int_0^h \mathbb{E}\|\nabla f(\widehat{\mathbf{x}}_n(\alpha)) - \nabla f(\mathbf{x}_n(s))\|^2 \mathrm{d}s.$$

By (C.3), we have $h^2 \int_0^h \frac{1}{h}\mathbb{E}\Omega_n(t)\mathrm{d}t = h^2 \mathbb{E}_{t\sim\mathrm{unif}[0,h]}\mathbb{E}\Omega_n(t) \lesssim h^2 \mathbb{E}_{t\sim\rho}\mathbb{E}\Omega_n(t)$.

When the Hessian of $f$ is upper bounded by $H$, by (D.6), we have

$$\mathbb{E}\|\mathbf{x}_n(t) - \mathbf{x}_n - \mathbf{x}_n^*(t) + \mathbf{x}_n^* + \mathbf{v}_n(t) - \mathbf{v}_n - \mathbf{v}_n^*(t) + \mathbf{v}_n^*\|^2$$

$$\lesssim h^2 \mathbb{E}_{t\sim\rho}\mathbb{E}\Omega_n(t) + u^2 h^2 \left( h^2 L^2 \mathbb{E}\Omega_n + \frac{h^2 L}{m}\mathrm{tr}(H) + L^2 h^2 \|\mathbf{x}_*\|^2 \right)$$

$$\lesssim h^2 \mathbb{E}\mathbb{E}_{t\sim\rho}\Omega_n(t) + h^4 \mathbb{E}\Omega_n + u^2 \frac{h^4 L}{m}\mathrm{tr}(H) + h^4 \|\mathbf{x}_*\|^2,$$

which completes the first claim.

When the $f$ is $L_2$-Hessian smooth, by (D.7), we have

$$\mathbb{E}\|\mathbf{x}_n(t) - \mathbf{x}_n - \mathbf{x}_n^*(t) + \mathbf{x}_n^* + \mathbf{v}_n(t) - \mathbf{v}_n - \mathbf{v}_n^*(t) + \mathbf{v}_n^*\|^2$$

$$\lesssim h^2 \mathbb{E}_{t\sim\rho}\mathbb{E}\Omega_n(t) + u^2 h^2 \left( h^2 L^2 \mathbb{E}\Omega_n + \frac{h^2 LM}{m} + L^2 h^2 \|\mathbf{x}_*\|^2 + L_2^2 u^2 h^6 d^2 \right)$$

$$\lesssim h^2 \mathbb{E}\mathbb{E}_{t\sim\rho}\Omega_n(t) + h^4 \mathbb{E}\Omega_n + u^2 \frac{h^4 LM}{m} + h^4 \|\mathbf{x}_*\|^2 + L_2^2 u^4 h^8 d^2,$$

which establishes the second claim.

$\square$

## D.3 Bound of $\mathbb{E}\|\mathbb{E}_{\beta\sim\rho'}\mathbb{E}_{\alpha\sim\rho'}\left(\nabla f(\mathbf{x}_n(\beta)) - \nabla f(\widehat{\mathbf{x}}_n(\alpha))\right)\|^2$

**Lemma D.5.** Let $\mathbf{x}_n(t)$ and $\widehat{\mathbf{x}}_n(t)$ be defined as above and $u = \frac{1}{L+m}$. Then,

(i) When Assumption 3.1 holds, and $H$ be defined as Definition 3.2. Then

$$\mathbb{E}\|\mathbb{E}_{\beta\sim\rho'}\mathbb{E}_{\alpha\sim\rho'}\left(\nabla f(\mathbf{x}_n(\beta)) - \nabla f(\widehat{\mathbf{x}}_n(\alpha))\right)\|^2 \lesssim h^6 L^2 \Omega_n + \frac{h^6 L}{m}\mathrm{tr}(H) + h^6 L^2 \|\mathbf{x}_*\|^2.$$

(ii) When Assumptions 3.1, E.1 and E.2 hold, we have

$$\mathbb{E}\|\mathbb{E}_{\beta\sim\rho'}\mathbb{E}_{\alpha\sim\rho'}\nabla\left(f(\mathbf{x}_n(\beta)) - \nabla f(\widehat{\mathbf{x}}_n(\alpha))\right)\|^2$$

$$\lesssim h^6 L^2 \mathbb{E}\Omega_n + \frac{h^6 L}{m}M + h^6 L^2 \|\mathbf{x}_*\|^2 + L_2^2 u^2 h^{10} d^2.$$

*Proof.* Note that $\mathbf{x}_n(t)$ depends on $\alpha$. First, we have

$$\mathbb{E}\|\mathbb{E}_{\beta\sim\rho'}\mathbb{E}_{\alpha\sim\rho'}\nabla\left(f(\mathbf{x}_n(\beta)) - \nabla f(\widehat{\mathbf{x}}_n(\alpha))\right)\|^2$$

$$=\mathbb{E}\|\mathbb{E}_{\alpha\sim\rho'}\left(\mathbb{E}_{\beta\sim\rho'}\nabla f(\mathbf{x}_n(\beta)) - \mathbb{E}_{\alpha\sim\rho'}\nabla f(\widehat{\mathbf{x}}_n(\alpha))\right)\|^2$$

$$=\mathbb{E}\|\mathbb{E}_{\alpha\sim\rho'}\mathbb{E}_{s\sim\rho'}\left(\nabla f(\mathbf{x}_n(s)) - \nabla f(\widehat{\mathbf{x}}_n(s))\right)\|^2$$

$$\leq\mathbb{E}\mathbb{E}_{\alpha\sim\rho'}\mathbb{E}_{s\sim\rho'}\|\nabla f(\mathbf{x}_n(s)) - \nabla f(\widehat{\mathbf{x}}_n(s))\|^2$$

$$=\mathbb{E}_{\alpha\sim\rho'}\mathbb{E}_{s\sim\rho'}\mathbb{E}\left[\|\nabla f(\mathbf{x}_n(s)) - \nabla f(\widehat{\mathbf{x}}_n(s))\|^2|\alpha, s\right].$$

Define $\bar{\mathcal{H}}'' = \int_0^1 \nabla^2 f(\mathbf{x}_n(s) + k(\widehat{\mathbf{x}}_n(s) - \mathbf{x}_n(s)))$ and recall that $\bar{\mathcal{H}}' = \int_0^1 \nabla^2 f(\widehat{\mathbf{x}}_n(\alpha) + k(\mathbf{x}_n - \widehat{\mathbf{x}}_n(\alpha)))\mathrm{d}k$. And then $\mathbb{E}\left[\|\nabla f(\mathbf{x}_n(s)) - \nabla f(\widehat{\mathbf{x}}_n(s))\|^2 \big| \alpha, s\right]$ can be bounded by

$$\mathbb{E}\left[\|\nabla f(\mathbf{x}_n(s)) - \nabla f(\widehat{\mathbf{x}}_n(s))\|^2 \big| \alpha, s\right]$$
$$= \mathbb{E}\left[\|\bar{\mathcal{H}}''(\mathbf{x}_n(s) - \widehat{\mathbf{x}}_n(s))\|^2 \big| \alpha, s\right]$$
$$\overset{\mathrm{a}}{=} \mathbb{E}\left[\left\|u\bar{\mathcal{H}}'' \int_0^s A_{12}(t-s)(\nabla f(\widehat{\mathbf{x}}_n(\alpha)) - \nabla f(\mathbf{x}_n))\mathrm{d}t\right\|^2 \Bigg| \alpha, s\right]$$
$$\overset{\mathrm{b}}{\leq} u^2 h^4 L^2 \mathbb{E}\left[\|\nabla f(\widehat{\mathbf{x}}_n(\alpha)) - \nabla f(\mathbf{x}_n)\|^2 \big| \alpha, s\right],$$

where $\overset{\mathrm{a}}{=}$ is by plugging in the close-form solution of $\mathbf{x}_n(s)$ and $\widehat{\mathbf{x}}_n(s)$; $\overset{\mathrm{b}}{\leq}$ follows by $A_{12}(t-s) \lesssim h$ and the Lipschitz continuous of $\nabla f$.

$$\mathbb{E}\|\mathbb{E}_{\beta\sim\rho'}\mathbb{E}_{\alpha\sim\rho'}\nabla\big(f(\mathbf{x}_n(\beta)) - \nabla f(\widehat{\mathbf{x}}_n(\alpha))\big)\|^2$$
$$\lesssim u^2 h^4 L^2 \mathbb{E}\|\nabla f(\widehat{\mathbf{x}}_n(\alpha)) - \nabla f(\mathbf{x}_n)\|^2$$
$$\overset{\mathrm{a}}{\lesssim} u^2 h^4 \left(h^2 L^4 \mathbb{E}\Omega_n + \frac{h^2 L^2}{m}\mathbb{E}\mathrm{tr}(\bar{\mathcal{H}}'^2) + h^2 L^4 \|\mathbf{x}_*\|^2 + uL^2\mathbb{E}\left\|\int_0^\alpha A_{12}(t-\alpha)\bar{\mathcal{H}}'\mathrm{d}\mathbf{B}_t\right\|^2\right)$$
$$\lesssim h^6 L^2 \mathbb{E}\Omega_n + \frac{h^6}{m}\mathbb{E}\mathrm{tr}(\bar{\mathcal{H}}'^2) + h^6 L^2 \|\mathbf{x}_*\|^2 + u^3 h^4 L^2 \mathbb{E}\left\|\int_0^\alpha A_{12}(t-\alpha)\bar{\mathcal{H}}'\mathrm{d}\mathbf{B}_t\right\|^2,$$

where $\overset{\mathrm{a}}{\lesssim}$ follows by (D.4). When $\nabla^2 f$ has a uniform upper bound $H$, we have

$$\mathbb{E}\|\mathbb{E}_{\beta\sim\rho'}\mathbb{E}_{\alpha\sim\rho'}\nabla\big(f(\mathbf{x}_n(\beta)) - \nabla f(\widehat{\mathbf{x}}_n(\alpha))\big)\|^2$$
$$\lesssim h^6 L^2 \mathbb{E}\Omega_n + \frac{h^6}{m}\mathbb{E}\mathrm{tr}(\bar{\mathcal{H}}'^2) + h^6 L^2 \|\mathbf{x}_*\|^2 + uh^7\mathrm{tr}(H^2)$$
$$\lesssim h^6 L^2 \mathbb{E}\Omega_n + \frac{h^6 L}{m}\mathrm{tr}(H) + h^6 L^2 \|\mathbf{x}_*\|^2,$$

where in the last inequality, we use that $\mathbb{E}\mathrm{tr}(\bar{\mathcal{H}}'^2) \leq L\mathbb{E}\mathrm{tr}(\bar{\mathcal{H}}') \leq L\mathrm{tr}(H)$ and $\mathrm{tr}(H^2) \leq L\mathrm{tr}(H)$.

And when Assumptions E.1 and E.2 hold, by Lemma D.3,

$$\mathbb{E}\|\mathbb{E}_{\beta\sim\rho'}\mathbb{E}_{\alpha\sim\rho'}\nabla\big(f(\mathbf{x}_n(\beta)) - \nabla f(\widehat{\mathbf{x}}_n(\alpha))\big)\|^2$$
$$\lesssim h^6 L^2 \mathbb{E}\Omega_n + \frac{h^6}{m}\mathbb{E}\mathrm{tr}(\bar{\mathcal{H}}'^2) + h^6 L^2 \|\mathbf{x}_*\|^2 + uh^7 LM + L_2^2 u^2 h^{10} d^2$$
$$\lesssim h^6 L^2 \mathbb{E}\Omega_n + \frac{h^6 L}{m}M + h^6 L^2 \|\mathbf{x}_*\|^2 + L_2^2 u^2 h^{10} d^2.$$

$\square$

# E  Local Hessian bound results

As discussed in Section 4.1, here we show that the uniform Hessian upper bound can be relaxed to local trace control with an additional Hessian smooth assumption. Specifically, we make the following assumptions on $f$.

**Assumption E.1.** $f$ has a $L_2$-Lipschitz Hessian.

Assumption E.1 is a standard assumption that frequently appears in the literature of optimization and sampling [Ma et al., 2021, Dalalyan and Riou-Durand, 2020, Durmus and Moulines, 2019].

**Assumption E.2.** There exist $M > 0$ such that for all $\mathbf{x} \in \mathbb{R}$, $\mathrm{tr}(\nabla^2 f(\mathbf{x})) \leq M$.

Assumption E.2 relax the uniform Hessian bound $\nabla^2 f(\mathbf{x}) \preceq H$ for $\mathbf{x} \in \mathbb{R}^d$ under which we obtain a convergence rate with $\mathrm{tr}(H)$. Assumption E.2 only requires that the trace of Hessian can be controlled locally. Under Assumptions E.1 and E.2, the convergence of DRUL only has a weak dimension dependency, as stated in Theorem E.3.

**Theorem E.3.** For any tolerance $\epsilon \in (0,1)$, denote the minimizer of $U(\mathbf{x})$ by $\mathbf{x}_*$. Assume that Assumptions 3.1, E.1 and E.2 hold. Set the step size

$$h \leq \min \left\{ \frac{1}{12C_2\kappa}, \frac{\epsilon^{2/3}}{\left(48C_2\kappa \left(\frac{u^2 L}{m} M + \|\mathbf{x}_*\|^2\right)\right)^{1/3}}, \frac{\epsilon^{2/7}}{\left(48C_2 L_2^2 u^4 d^2\right)^{1/7}} \right\},$$

where $C_2 \geq 1$ is a universal constant. With initial point $(\mathbf{x}_0, \mathbf{v}_0)$, define $\Omega_0 = \mathbb{E}_{\mathbf{x}\sim p, \mathbf{v}\sim\mathcal{N}(0,u)} \left( \|\mathbf{x}_0 - \mathbf{x}\|^2 + \|\mathbf{x}_0 + \mathbf{v}_0 - \mathbf{v} - \mathbf{x}\|^2 \right)$. Then under Assumptions 3.1, when

$$n \geq \frac{8e\kappa}{h} \log\left(\frac{2\mathbb{E}\Omega_0}{\epsilon^2}\right),$$

Algorithm 1 outputs $\mathbf{x}_n$ such that $W_2(\mathrm{Law}(\mathbf{x}_n), p) \leq \epsilon$.

Without loss of generality, assume $\mathbf{x}_* = 0$. Then Theorem E.3 indicates a convergence rate of $\widetilde{\mathcal{O}}\left(M^{1/3}\epsilon^{-2/3} + d^{2/7}\epsilon^{-2/7}\right)$, which has a lower dimension dependency both in the sense of the order of $d$ and the relevant $\epsilon$ coefficient. When the Hessian of any $\mathbf{x} \in \mathbb{R}^d$ has a dimension-free trace, we arrive at a convergence rate of $\mathcal{O}\left(\epsilon^{-2/3} + d^{2/7}\epsilon^{-2/7}\right)$.

### E.1 Proof of Theorem E.3

In this section, we prove the convergence guarantee using the upper bound of the local Hessian trace.

*Proof.* We follow the notations in the proof of Theorem 4.2 in Appendix B.2. Similarly, denote the error in (B.5) by

$$A = 2uhC_1\mathbb{E}\mathbb{E}_{\beta\sim\rho'}\langle \mathbf{x}_n(\beta) - \mathbf{x}_n^*(\beta) + \mathbf{v}_n(\beta) - \mathbf{v}_n^*(\beta), \nabla f(\mathbf{x}_n(\beta)) - \nabla f(\widehat{\mathbf{x}}_n(\alpha))\rangle.$$

Then

$$
\begin{aligned}
A \leq & 2C_1 uh\mathbb{E}\langle \mathbf{x}_n(\beta) - \mathbf{x}_n - \mathbf{x}_n^*(\beta) + \mathbf{x}_n^* + \mathbf{v}_n(\beta) - \mathbf{v}_n - \mathbf{v}_n^*(\beta) + \mathbf{v}_n^*, \nabla f(\mathbf{x}_n(\beta)) - \nabla f(\widehat{\mathbf{x}}_n(\alpha))\rangle \\
& + 2C_1 uh\mathbb{E}\langle \mathbf{x}_n - \mathbf{x}_n^* + \mathbf{v}_n - \mathbf{v}_n^*, \mathbb{E}_{\alpha\sim\rho'}\mathbb{E}_{\beta\sim\rho'}\left(\nabla f(\mathbf{x}_n(\beta)) - \nabla f(\widehat{\mathbf{x}}_n(\alpha))\right)\rangle \\
\lesssim & \frac{uh}{uh}\mathbb{E}\|\mathbf{x}_n(\beta) - \mathbf{x}_n - \mathbf{x}_n^*(\beta) + \mathbf{x}_n^* + \mathbf{v}_n(\beta) - \mathbf{v}_n - \mathbf{v}_n^*(\beta) + \mathbf{v}_n^*\|^2 \\
& + u^2 h^2 \mathbb{E}\|\nabla f(\mathbf{x}_n(\beta)) - \nabla f(\widehat{\mathbf{x}}_n(\alpha))\|^2 + \frac{h}{u}uh\mathbb{E}\|\mathbf{x}_n - \mathbf{x}_n^* + \mathbf{v}_n - \mathbf{v}_n^*\|^2 \\
& + \frac{u}{h}uh\mathbb{E}\|\mathbb{E}_{\alpha\sim\rho'}\mathbb{E}_{\beta\sim\rho'}\left(\nabla f(\mathbf{x}_n(\beta)) - \nabla f(\widehat{\mathbf{x}}_n(\alpha))\right)\|^2 \\
\overset{a}{\lesssim} & \left( h^2\mathbb{E}\mathbb{E}_{t\sim\rho}\Omega_n(t) + h^4\mathbb{E}\Omega_n + u^2\frac{h^4 LM}{m} + h^4\|\mathbf{x}_*\|^2 + L_2^2 u^4 h^8 d^2 \right) \\
& + u^2 h^2 \left( h^2 L^2\mathbb{E}\Omega_n + \frac{h^2 L}{m}M + L^2 h^2\|\mathbf{x}_*\|^2 + L_2^2 u^2 h^6 d^2 \right) + h^2\mathbb{E}\Omega_n \\
& + u^2 \left( h^6 L^2\mathbb{E}\Omega_n + \frac{h^6 L}{m}M + h^6 L^2\|\mathbf{x}_*\|^2 + L_2^2 u^2 h^{10} d^2 \right) \\
\lesssim & h^2\mathbb{E}\mathbb{E}_{\beta\sim\rho}\Omega_n(\beta) + h^2\mathbb{E}\Omega_n + \frac{u^2 h^4 L}{m}M + h^4\|\mathbf{x}_*\|^2 + L_2^2 u^4 h^8 d^2,
\end{aligned}
$$

where $\overset{a}{\lesssim}$ follows by Lemmas D.1, D.4 and D.5. We shall note that $\mathbb{E}\Omega_{n+1} = \mathbb{E}_{\beta\sim\rho}\Omega_n(\beta)$. Combining the above control with (B.5), we have

$$
\begin{aligned}
\mathbb{E}\Omega_{n+1} \leq & \frac{\mathbb{E}_{\beta\sim\rho}e^{-\frac{\beta}{\kappa}} + C_2 h^2}{1 - C_2 h^2}\mathbb{E}\Omega_n + \frac{C_2 h^4}{1 - C_2 h^2}\left(\frac{u^2 L}{m}M + \|\mathbf{x}_*\|^2 + L_2^2 u^4 h^4 d^2\right) \\
\leq & \left(1 - \frac{h}{6\kappa}\right)\mathbb{E}\Omega_n + 2C_2 h^4\left(\frac{u^2 L}{m}M + \|\mathbf{x}_*\|^2 + L_2^2 u^4 h^4 d^2\right),
\end{aligned}
$$

where the last inequality follows by the control (B.8) and the choice of $h$ which satisfies $1 - C_2 h^2 \geq \frac{1}{2}$. Denote $r = 1 - \frac{h}{6\kappa}$ and $\mathcal{E}' = 2C_2 h^4 \left( \frac{u^2 L}{m} M + \|\mathbf{x}_*\|^2 + L_2^2 u^4 h^4 d^2 \right)$ and therefore

$$\mathbb{E}\Omega_N \leq r\mathbb{E}\Omega_{N-1} + \mathcal{E} \leq r^N \mathbb{E}\Omega_0 + \mathcal{E}(1 + r + \cdots + r^{N-1}) \leq r^N \mathbb{E}\Omega_0 + \frac{\mathcal{E}}{1 - r}.$$

When $N \geq \frac{6}{\kappa} \log \left( \frac{2\mathbb{E}\Omega_0}{\epsilon^2} \right)$, we have

$$r^N \mathbb{E}\Omega_0 \leq e^{\frac{Nh}{6\kappa}} \mathbb{E}\Omega_0 \leq \frac{\epsilon^2}{2}. \tag{E.1}$$

And when

$$h \leq \min \left\{ \frac{\epsilon^{2/3}}{\left( 48C_2\kappa \left( \frac{u^2 L}{m} M + \|\mathbf{x}_*\|^2 \right) \right)^{1/3}}, \frac{\epsilon^{2/7}}{\left( 48C_2 L_2^2 u^4 d^2 \right)^{1/7}} \right\},$$

we get $\frac{\mathcal{E}}{1-r} \leq \frac{\epsilon^2}{2}$. Together with the (E.1), we obtain the convergence rate in Theorem E.3. $\qquad \square$

Finally, we would say that the trace of Hessian should be a more realistic factor to characterize the complexity of Langevin algorithms. Beyond the ridge separable functions described in Section 3.3, we illustrate the Bayesian sampling over the two-layer neural network model and show that the neural network also has a bounded trace under suitable conditions. Note that our convergence analysis cannot be applied to this task since neural networks are non-convex models, and we leave the extensions of analysis to the general non-log-concave case as future work.

**Proposition E.4.** Define $f(\mathbf{W}, \mathbf{w}) = \mathbf{w}^\top \sigma(\mathbf{W}^\top \mathbf{x})$, where $\sigma$ is the activation function. When $\|\mathbf{x}\|_1 \leq r_1$, $\|\mathbf{w}\| \leq r_2$ and $\sigma''(x) \leq \alpha$, we have $\mathrm{tr}\left( \nabla^2 f(\mathbf{W}, \mathbf{w}) \right) \leq \alpha r_1 r_2$.

*Proof.* By direct computation, we have

$$\frac{\partial f}{\partial \mathbf{w}} = \sigma(\mathbf{W}^\top \mathbf{x}), \qquad \qquad \frac{\partial f}{\partial \mathbf{W}} = \left( \sigma'(\mathbf{W}^\top \mathbf{x}) \odot \mathbf{w} \right) \otimes \mathbf{x},$$

$$\frac{\partial^2 f}{\partial \mathbf{w}^2} = \mathbf{0}, \qquad \qquad \frac{\partial^2 f}{\partial \mathbf{W}^2} = \mathrm{Diag}(\sigma''(\mathbf{W}^\top \mathbf{x}) \odot \mathbf{w}) \otimes \mathbf{x} \otimes \mathbf{x}.$$

Therefore,

$$\begin{aligned}
\mathrm{tr}\left( \nabla^2 f(\mathbf{W}, \mathbf{w}) \right) &= \|\mathbf{x}\|^2 \cdot \mathrm{tr}\left( \mathrm{Diag}(\sigma''(\mathbf{W}^\top \mathbf{x}) \odot \mathbf{w}) \right) \\
&\leq r_1^2 \cdot \langle \sigma''(\mathbf{W}^\top \mathbf{x}), \mathbf{x} \rangle \\
&\leq \alpha r_1 r_2.
\end{aligned} \tag{E.2}$$

$\square$

