# OpenReview forum: "Double Randomized Underdamped Langevin with Dimension-Independent Convergence Guarantee"
_NeurIPS.cc/2023/Conference — NeurIPS 2023 poster_

### Official Review · Reviewer_RFtN · 2023-07-02

**Soundness:** 3 good
**Presentation:** 3 good
**Contribution:** 3 good
**Rating:** 7
**Confidence:** 3

**Summary:**

This paper considers the problem of sampling from a Gibbs distribution $p(x) \propto e^{-U(x)}$ using discretized Langevin dynamics. Since the approximation error of such methods usually depends on $\mathrm{Tr}(\nabla^2 U)$, the proposed algorithm splits $U$ into a quadratic part $g(x) = \frac m2 ||x||^2$ and a remainder $f$, and only discretizes the dynamics according to $f$.

Such a split was already considered in [Freund et al. '21]; however, the main novelty of this article is that the discretization on $f$ is performed through a two-step method with random step-sizes, instead of a simple gradient update. This scheme shaves off a factor of $\epsilon^{-1/3}$ from the required time complexity to reach a Wasserstein error of $\epsilon$.

The proof is based on an approximate contraction bound for a quantity $\Omega_n$ that bounds the desired Wasserstein distance, followed by a precise analysis and optimization of the error in the aforementioned bound.

**Strengths:**

This paper presents a novel algorithm for Langevin simulation, that achieves state-of-the-art performance for the dependency on both the precision requirement $\epsilon$ and the ambient dimension $d$. The proposed algorithm is fairly simple (at least for quadratic $g$) and easy to implement, and the main ideas behind it are clearly explained. Overall, the paper is fairly well-written, with only a few typos here and there.

**Weaknesses:**

Clarity/soundness: some of the proofs are very hard to parse due to the amount of simplifications made at once from one line to the next. This is especially felt in B.2, where the first two inequalities below l.447 contain around 5 sub-inequalities to check each. The specifications $\alpha \sim \rho'$ and $\beta \sim \rho$ are also used inconsistently, which makes it hard to understand with respect to what quantities each expectation is taken.

Novelty/significance: The relationship to [Shen and Lee '19] and [Freund et al. '22] should probably be expanded. From what I understand, this paper unites the randomized midpoint method of the former with the combined optimization viewpoint of the latter, but it is unclear if there are challenges other than computational to this endeavor. Namely, neither of those methods require such a complicated step-size scheme, which seems to be the main novelty of the paper, but the need for it is unclear.

Minor remarks:
- eq. (3.5): what is A?
- l.185: "squre"
- the equation below l.279 should be a scalar product.
- in (B.4), shouldn't $z_n(t)$ be $\hat x_n(\alpha) - x_n^*(t)$ instead ? This change does ripple through the proof of Lemma 5.1, so I'm not actually sure of how minor it is.

**Questions:**

- How important is the choice of $\rho$ ? I understand from Lemma 5.2 and its implications that for a given $\rho$, the choice of $\rho'$ is important to ensure these conditions, but why can't I, for example, choose a uniform prior for $\beta$?
- In section 5.2, it seems like you take $\bar w_n = x_n - x_n^* + v_n - v_n^*$ as the expectation of $w_n(s, \alpha)$; why is it the case ?

---

> ### Author Rebuttal · Authors · 2023-08-10
>
> ● Neither of those methods require such a complicated step-size scheme, which seems to be the main novelty of the paper, but the need for it is unclear.
>
>
> We introduce the random step size to bound the error such as $\|\int_0^t x_n(s) - x_n^*(t)\mathrm{d} s\|^2$ in the discretization analysis. The $\|\int_0^t x_n(s) - x_n^*(t)\mathrm{d} s\|^2$ may be dimension-dependent as a standard bound will lead to a $dt$ additional discretization error. We seek to bound this kind of error using the descent on $\Omega_n(t)$. Under the stochastic step size, $\Omega_{n+1}$ will be $\mathbb{E} \Omega_n(t)$, and thus we can bound the dimension-dependent discretization error as $\|\int_0^t x_n(s) - x_n^*(t)\mathrm{d} s\|^2\lesssim \mathbb{E}\Omega_{n}(t)$. And then there reminds additional conditions on $\rho$ and $\rho'$ to
>
> ● The specifications $\alpha\sim\rho'$ and $\beta\sim\rho$ are also used inconsistently, which makes it hard to understand with respect to what quantities each expectation is taken.
>
> Thanks for pointing out the clearness issue in the presentation. We frequently use Claim (B) in our proof which indicates that we can replace the expectation under $\rho’$ with the one under $\rho$ up to a constant level. Mostly, its purpose is to control the error term with the expectation of $\rho'$ using the contraction which is an expected value over $\rho$. And we use Claim (A) in Lemma D.3. We will make more explanations for our proof especially on the specification of the $\rho$ and $\rho'$ in a further revised version.
>
>
> ● How important is the choice of $\rho$?
>
> We believe that $\rho$ is not restrictive to the proposed one. However, this choice of $\rho$ has to satisfy several conditions. Our choice of $\rho$ induces a simple $\rho'$ and is a natural choice. We require its expectation is of $\Theta(h)$ order which guarantees sufficient descent and the average descent can control the dimension-dependent error as in (D.18). And the corresponding $\rho’$ should satisfy claim (B) in Lemma 5.2. Uniform $\rho$ is possible given that corresponding $\rho’$ is chosen accordingly by Lemma 5.2. But this requires further validations and induces a more complicated $\rho'$. As a reminder, uniform $\rho$ does not lead to the random step size version of the randomized midpoint method. The distribution of $\rho’$ of the stochastic step size randomized midpoint method is different and it does not satisfy claim (B) in Lemma 5.2.
>
>
>
> ● in (B.4), shouldn’t $z_n(t)$ be $\hat{x}_n(\alpha) - x_n^*(t)$.
>
> We appreciate the careful revision and pointing out this issue. This is our typo and does not influence the following analysis. This typo can be fixed only by changing the equation in line 430 and line 431. $z_n(t)$ is $x_n(t) - x_n^*(t)$. But Equation (B.4) should be $\nabla U(x_n(t)) - U(x_n^*(t)) = \int_0^1 \nabla^2 U(s\hat{x}_n(\alpha) + (1-s)x_n^*(t)) \mathrm{d} s z_n(t)$. And the following analysis remains unchanged. The last inequality follows by dividing $\nabla f(\hat{x}_n(\alpha)) - \nabla f(x_n^*(t))$ into $\nabla f(\hat{x}_n(\alpha)) - \nabla f(x_n(t))$ and $ f(\hat{x}_n(\alpha)) - \nabla f(x_n^*(t))$.
>
>
> ● In section 5.2, it seems like you take $\bar{w}_n= x_n - x_n^* + v_n - v_n^*$ as the expectation of $w_n(s,\alpha)$.
>
> Thanks for pointing out the unclearness in our presentation. We do not take $\bar{w}$ as the expectation of $w_n(s,\alpha)$. There is a higher-order discretization error gap between the approximation $\bar{w}$ and $w_n(s,\alpha)$. Here we illustrate that claim (A) of 5.2 leads to a randomized mid-point discretization. One can refer to the first equation in line 447 for the rigorous proof. And we will specify the error term and our choice of $\bar{w}$ in the main text in a later refined version.

---

> ### Comment · Reviewer_RFtN · 2023-08-16
>
> Thank you for your response. While the paper still might need a clarity pass, I am now more convinced of the technical novelty of introducing this double randomization scheme. I have therefore raised my score.

---

### Official Review · Reviewer_5mo9 · 2023-07-06

**Soundness:** 3 good
**Presentation:** 3 good
**Contribution:** 3 good
**Rating:** 6
**Confidence:** 4

**Summary:**

The paper suggests a novel version of the Unadjusted Langevin Algorithm with sample complexity in Wasserstein-2 distance scaling with the effective dimension of the problem (trace of the potential's Hessian) instead of the ambient space dimension in case of strongly log-concave distributions. This result completes and generalizes the results of [Shen and Lee, 2019].

**Strengths:**

The research direction towards studying the sample complexity rates in terms of the effective dimension is interesting and potentially allows to explain the successful behaviour of the ULA-type algorithms in the high-dimensional problems.

**Weaknesses:**

First of all, the suggested Algorithm 1 (DRUL) does not seem to be really an implementable one, since the discretization error in $x_{n+1}$ and $v_{n+1}$ is to appear in the practical implementations. It is not clear if the control of this discretization error would not yield an explicit dimension dependence in the stepsize $h$ in Theorem 4.2.

Second, the sample complexity scaling as $\varepsilon^{-2/3}$ is not completely convincing. For example, for the ridge separable potentials, which are the main motivating example towards the paper, the Hamiltonian Monte-Carlo method is known to obtain a sample complexity of order $(d/\varepsilon)^{1/4}$, see e.g. [Mangoubi et al, 2017]. Thus there is a natural question if the $\varepsilon^{-2/3}$ complexity optimal for Langevin-type algorithms? Again it seems that the particular rate can degrade after the intergral discretization in Alg. 1 taken into account.

References:
Mangoubi, O., & Smith, A. (2017). Rapid mixing of Hamiltonian Monte Carlo on strongly log-concave distributions. arXiv preprint arXiv:1708.07114.

**Questions:**

1. Is it possible to add any numerical findings illustrating the superiority of the doubly randomized ULA (with random step size) against the one with constant or decreasing step size? If one could trace the precise dependence upon the $\operatorname{trace}{H}$ even in the toyish setup, I would lean towards increasing my score.

2. Are there any novel technical contributions developed to prove the result of Theorem 4.2? If yes, please add the corresponding discussion to the main text.

**Limitations:**

The paper is theoretical and no negative societal impact is expected.

---

> ### Author Rebuttal · Authors · 2023-08-10
>
> ● First of all, the suggested Algorithm 1 (DRUL) does not seem to be really an implementable one, […] It is not clear if the control of this discretization error would not yield an explicit dimension dependence in the stepsize $h$ in Theorem 4.2.
>
> We apologize for the possible misunderstanding in the presentation of ALgorithm 1. We would like to clarify that Algorithm 1 is implementable and there is no dimension dependence error or necessity of further discretization. The discretized process (4.1) is linear in $x_n(t)$ and $v_n(t)$ and thus closed form solution is given by a Gaussian distribution. The exact solution is given by (4.3). Also note that the integral in Algorithm 1 is tractable, e.g. $\int_0^{\alpha_n} A_{12}(s-\alpha_n)\nabla f(x_n)\mathrm{d}s = \int_0^{\alpha_n} A_{12}(s-\alpha_n)\mathrm{d}s \nabla f(x_n)$ and $A_{12}$ defined in Lemma 4.1 has closed-from integral. To implement Algorithm 1, one only needs to sample from a Gaussian distribution whose mean is tractable and the variance matrix is given by (A.3). We will specify the solution of the integral in Algorithm 1 and the covariance in (A.3) in a later revised version.
>
> ● Second, the sample complexity scaling as $\epsilon^{-2/3}$ is not completely convincing. […] can degrade after the integral discretization in Alg. 1 is taken into account.
>
> We would like to clarify that the ridge separable case is an illustration that $\mathrm{tr}(H)$ can be dimension independent. The analysis is not restricted to the ridge separable case and reduces the dimension dependence for a wide range of problems that have rapidly dropping Hessian eigenvalues. One can refer to our discussion at the end of section 4.1.
>
> And Mangoubi et al (2017) do not assume the ridge separable structure. They assume in assumption 1.7 that the potential $U(x)$ can be separated according to the blocked state space, that is $U(x) = \sum_{i=1}^{d/m}U(x_{i})$ where $x_i\in\mathbb{R}^{m}$. Although Mangoubi et al (2017) prove that HMC achieves $d^{1/4}$ or $\epsilon^{1/4}$ rate, the convergence guarantees require higher-order discretization schemes. These higher-order ODE solvers require higher-order smoothness which is not necessary in our algorithm. Such as the leap-frog scheme requires third-order smoothness to obtain the guarantee.
>
> $\left(\frac{d}{\epsilon}\right)^4$ can be achieved via higher-order Langevin schemes by Mou et al. (2021) (High-order Langevin diffusion yields an accelerated MCMC algorithm) for ridge separable case. However, it is based on a different oracle $\Delta U$ other than the gradient oracle considered in our algorithms. And the convergence has a much worse dependence on conditioning number. The difference oracle $\Delta U$ is usually intractable. For general target distributions, they require numerical integration and the convergence rate degrades.
>
> As for the optimality, the discretization scheme with convergence rate $\epsilon^{-2/3}$ is known to be optimal for underdamped Langevin algorithms [Cao et al. (2020)] (Complexity of randomized algorithms for underdamped Langevin dynamics). It is a discretization lower bound.
>
> ● Is it possible to add any numerical findings illustrating the superiority of the doubly randomized ULA (with random step size) against the one with constant or decreasing step size?
>
> We have conducted some synthetic studies to show that our DRUL (with random step size) can achieve better mixing distribution compared to the randomized midpoint with a constant step size. Here we illustrate the dimension dependence using synthetic data. We consider Bayesian ridge regression with $g(w) = 0.1\|w\|^2/2$ and $f(w) =\frac{1}{2n} \|Xw - Y\|^2$, where rows of $X$ have unit norm. $f(x)$ has dimension-independent trace. We evaluate the result by a projection to one dimension and calculate the empirical Wasserstein distance (since it is non-trivial to obtain precise high dimensional distance using empirical data). We fix the step size and compare the mixing result. We take the average of 3 independent evaluations. Since the algorithms achieve high precision, we require a large number of samples (500000) to get an accurate estimate of Wasserstein distance even in one dimension. It is a huge computation cost in high-dimension regime. Thus we fixed a large step $h=1$ (under which the mixing step is low) size and $T = 300$ steps to guarantee mixing. Below is the results, where we compare the result of the randomized midpoint method (RMM) and our DRUL method when the dimension $d = 5,10,100,1000$.
>
> | Dimension | 5      | 10     | 100    | 1000   |
> |-----------|--------|--------|--------|--------|
> | RMM       | 0.0084 | 0.0083 | 0.0180 | 0.0416 |
> | DRUL      | 0.0085 | 0.0071 | 0.0050 | 0.0077 |
>
>
> From the table, DURL achieves more accurate mixing distributions in high-dimension space, thus showing superiority as the dimension grows. Note that by directly sampling from the target distribution 500000 times, the empirically estimated distance from the target distribution is around 0.001-0.005. DRUL method may achieve better results than the ones shown in the table.
>
> As a reminder, our work is not a stochastic step size Midpoint method. The stochastic step size Midpoint method will induce a differential $\rho’$ that does not satisfies claim (B) in Lemma 5.2. Besides, the decreasing stepsize also applies to our method.
>
> ● Are there any novel technical contributions developed to prove the result of Theorem 4.2? If yes, please add the corresponding discussion to the main text.
>
> We develop a new discretization scheme and bound the error-dependent term by an averaged effect as discussed in Section 5. The analysis requires delicate control and devising on the choice of $\rho$ and $\rho'$. And our analysis bounds the local error in the process, which leads to our choice of $\rho$ and $\rho'$. We will make a more clear and thorough discussion in a revised version.

---

> > ### Comment · Reviewer_5mo9 · 2023-08-19
> >
> > Thank you for the detailed answer. Now I am more convinced with the theoretical contributions of the paper and I would raise my score.

---

### Official Review · Reviewer_SZ7b · 2023-07-07

**Soundness:** 4 excellent
**Presentation:** 4 excellent
**Contribution:** 2 fair
**Rating:** 5
**Confidence:** 4

**Summary:**

The paper adapts the Randomized Midpoint Method to the composite optimization context considered in Freund et al, and consequently improves the dependence from $O(tr(H)/\epsilon)$ to $O((tr(H)/\epsilon)^{1/3})$.

**Strengths:**

The application of randomized midpoint in this composite sampling is novel, and there is genuine improvement in the rate estimate when compared to Freund et al.

The technique of double randomization is new and requires some novel analysis when contrasted with prior works.

The authors do a decent job of illustrating that the trace of the Hessian is $o(d)$ through some figures and discussion.


**Weaknesses:**

The primary contributions of this paper are not particularly original and mostly stem from combining the framework in Freund et al. with the known analysis for randomized midpoint in Shen and Lee. This in my view is the primary weakness of the paper.

In general, claims about the “dimension-free” nature of the convergence guarantees need to be careful since the composite structure assumption is quite strong, although the authors in general do a good job of qualifying their claims.

Overall, while this work contains some novel claims and results, and is a bona fide improvement on prior work. However, the technical novelty is not significant, and I am borderline on this paper as a result.


**Questions:**

It seems inappropriate to compare this to the original randomized midpoint work/other work for the standard Langevin Monte Carlo, since they do not assume the composite structure of the problem. The primary highlighted comparison is with respect to Freund et al. (2022), in which case the gain is more like $tr(H)^{1/3}/\epsilon^{4/3}$. If $tr(H)$ is $O(1)$ then this is only a gain in epsilon, which is usually smaller than $d$.

What is the previous proof referred to in L. 253?

Terminology of “acceleration” should probably be avoided since this is classically used only to refer to sqrt(kappa) rates. A better term might be “improved discretization error”.

Typos:

L. 140 What is being made strongly convex?

L. 185 squred -> squared

L. 197 denote solution -> denote the solution


**Limitations:**

I have outlined my concerns already in the previous sections.

---

> ### Author Rebuttal · Authors · 2023-08-10
>
> ● The primary contributions of this paper are not particularly original and mostly stem from combining the framework in Freund et al. with the known analysis for the randomized midpoint in Shen and Lee. This in my view is the primary weakness of the paper.
>
> We would like to emphasize that our method is not a simple combination of the previous work.
>
> First, the two-stage optimization in Freund et al. cannot be directly extended to the accelerated process or complicated discretization scheme due to its form. Meanwhile, the midpoint discretization analysis introduces dimension-dependent errors such as in line 447, which does not exist in the analysis in the EU discretization of the overdamped Langevin analysis and underdamped analysis. This requires refined discretization schemes to eliminate the dimension-dependent error and thus there are underlying difficulties to extend to complicated discretization schemes. We design a discretization scheme and prove the scheme can achieve a convergence rate with $\mathrm{tr}(H)$ dependence. And this can be achieved for methods based on both accelerated processes and complicated discretizations. This implies that a wide range of Langevin algorithms can be adapted to achieve low dimension dependence.
>
>
> ● In general, claims about the “dimension-free” nature of the convergence guarantees need to be careful since the composite structure assumption is quite strong, although the authors in general do a good job of qualifying their claims.
>
> Sorry for the ambiguity of our statement. We would like to clarify that our analysis is not restricted to composite structure and applies to general strongly convex target distributions. As discussed in section 3.2, the composite structure also includes the general $m$-strongly convex function, which can be divided into $g(x) = \frac{m}{2} \|x\|^2$ and the weakly convex function $f (x) = U (x) −\frac{m}{2} \|x\|^2$. This leads to the same $\frac{\mathrm{tr}(H)^{1/3}}{\epsilon^{2/3}}$ convergence rate. One of the purposes of introducing composite structure in the proof is to alleviate the dimension dependence in the Hessian trace. The composite structure does not serve as the prerequisite of the analysis. Otherwise, the $\mathrm{tr}(H)$ will have an extra $md$ term. The scaling of $md$ is problem specific and may be dimension dependent when $\frac{1}{m} = o(d)$. The natural composite structure is ubiquitous in sampling tasks.
>
> ● It seems inappropriate to compare this to the original randomized midpoint work/other work for the standard Langevin Monte Carlo, since they do not assume the composite structure of the problem.
>
> ● L. 140 What is being made strongly convex?
>
> As discussed above, the algorithm applies to the general strongly convex distributions. $U$ is strongly convex. We will discuss the composite structure more in a further revised version.
>
> ● What is the previous proof referred to in L. 253?
>
> The proof in Shen and Lee (2019). We follow the seminal contraction-based method by Shen and Lee (2019) and Cheng et al. (2018). This is to illustrate the difference of the proof to one familiar with the proof by Shen and Lee. Our analysis tracks the difference between the implemented process and the exact process with the initial distribution corresponding target distribution, and considers to bound the local error in the process. It is consistent with the proof based on the gradient flow.
>
>
> ● Terminology of “acceleration” should probably be avoided since this is classically used only to refer to sqrt(kappa) rates.
>
> Thanks for pointing out this issue. Acceleration is indeed improper.

---

> > ### Comment · Reviewer_SZ7b · 2023-08-14
> > **Response**
> >
> > I thank the authors for their detailed response. Having read through the comments by the other reviewers and the subsequent discussion, I remain borderline since I am still skeptical that this paper presents a sufficiently novel contribution to merit acceptance. However, there does seem to be some genuine analytical novelty arising from the analysis of the discretization error, and I have raised my score by one point as a result.

---

> > > ### Author Response · Authors · 2023-08-16
> > > **Thank you**
> > >
> > > Thanks again for the suggestions and the recognition of our work. Please let us know if you have more questions or comments.

---

### Official Review · Reviewer_TP8b · 2023-07-18

**Soundness:** 3 good
**Presentation:** 2 fair
**Contribution:** 3 good
**Rating:** 6
**Confidence:** 2

**Summary:**

In this paper, the authors propose a Langevin-type algorithm for sampling a strongly log-concave distribution with a composite structure. Their method can be viewed as a variant of the randomized midpoint method, with two key modifications: (i) they only discretize the smooth convex part of the negative log likelihood but retain the strongly convex part; (ii) they draw both step sizes in the algorithm randomly according to carefully crafted distributions. It is shown that the algorithm achieves an accelerated rate without explicit dependence on the dimension.

**Strengths:**

- The result is interesting and noteworthy. To sample a strongly log-concave distribution, the best-known iteration complexity bound either has an undesirable dimension dependence, such as $\tilde{O}(\frac{d^{1/3}}{\epsilon^{2/3}})$ in [Shen and Lee, 2019], or a worse dependence on $\epsilon$, such as $\tilde{O}(\frac{\mathrm{tr}(H)}{\epsilon})$ in [Freund et al., 2022]. In this work, the authors manage to achieve the best of both worlds and prove a complexity bound of $\tilde{O}(\frac{(\mathrm{tr}(H))^{1/3}}{\epsilon^{2/3}})$.
- The authors introduce the double randomized technique to reduce the discretization error, which seems novel to me.

**Weaknesses:**

I think the presentation of the paper can be improved.

- In particular, the explanation in Section 5 is not very helpful, and it remains unclear to me why the randomized step size helps reduce the discretization error, and why the authors choose the specific distribution in Lemma 5.2. Moreover, it would be helpful if the authors can compare their analysis with the one in [Shen and Lee, 2019] to better explain how they remove the dimension dependence.

-  Also, there are numerous typos in the main text and the proofs in the appendix, which sometimes make it hard to understand. Please see the "Questions" section for more details.

**Questions:**

- While this work focuses on the dimension dependence, the condition number $\kappa = L/\mu$ can also impact the convergence rate greatly. How is the result in this paper compared with the existing works in terms of the dependence on $\kappa$?
- In Lemma 5.1, it is unclear to me what it means that "$x_n$ and $x_n^*$ are coupled synchronously". Do you mean that they are driven by the same Brownian process?
- Page 9, Lines 277-281: I am confused by this paragraph. By definition, isn't the random weight $w_n(s,\alpha)$ a $d$-dimension vector? If so, how can you apply Claim (A) in Lemma 5.2?
- Page 9, Lines 282-292: It is also unclear to me why "the randomized step size make it possible to consider the averaged effect".
- Page 12, (A.1): I am not sure why the authors introduce the extra parameter $\gamma$. As far as I can see, $\gamma$ is fixed as 2 in the rest of the proof.
- Page 14, Section B.1.1: I don't see why the random process $B_t^{\alpha}$ is a Brownian bridge. Is it supposed to be the random process $B_t$ conditioned on $B_{\alpha}$? And why do we need to introduce this process in the first place?
- Page 14, the equations under Line 431: Here the authors exchange the order of differentiation and expectation, which is not justified. Indeed, it is not even clear if $\Omega(t)$ is differentiable since it involves the solution trajectories of SDEs.


Typos in the paper:

- The convergence rates reported in the introduction are inconsistent with the ones in Table 1. Specifically, the rate by [Shen and Lee, 2019] should be $\tilde{O}(\frac{d^{1/3}}{\epsilon^{2/3}})$ (Page 2, Line 47), and the rate by [Freund et al., 2022] should scale linearly with $O(\frac{1}{\epsilon})$ (Table 1, Row 5).
- Page 1, Line 49: "convergence dependence" -> "dimension dependence"
- Page 5, Definition 3.6: $A$ is undefined.
- Page 9, Line 275: the integral should be $\int_{0}^t e^{\frac{s-t}{\kappa}} F(s) ds$.

**Limitations:**

Yes, the authors addressed the limitations of their work.

---

> ### Author Rebuttal · Authors · 2023-08-10
>
>  Page 9, Lines 282-292: It is also unclear to me why "the randomized step size makes it possible to consider the averaged effect".
> ● In particular, the explanation in Section 5 is not very helpful [...] better explain how they remove the dimension dependence.
>
> In short, the introduction of the random step size is to bound the error such as $\|\int_0^t x_n(s) - x_n^*(t)\mathrm{d} s\|^2$ in the discretization analysis. The $\|\int_0^t x_n(s) - x_n^*(s)\mathrm{d} s\|^2$ may be dimension-dependent as a standard bound will lead to a $dt$ additional discretization error. We seek to bound this kind of error using the descent on $\Omega_n(t)$. Under the stochastic step size, $\Omega_{n+1}$ will be $ \mathbb{E} \Omega_n(t) $,  and thus we can bound the dimension-dependent discretization error as $\|\int_0^t x_n(s) - x_n^*(s)\mathrm{d} s\|^2 \lesssim \mathbb{E}\Omega_{n}(t)$. And it remains a careful analysis to obtain the dimension-free error and delicate device of $\rho, \rho'$.
>
>
> ● While this work focuses on dimension dependence, the condition number $\kappa=L/μ$ can also impact the convergence rate greatly. How is the result in this paper compared with the existing works in terms of the dependence on $\kappa$?
>
> Thanks for your suggestions. This paper mainly considers dimension dependence. It is very interesting to study the dependence on $\kappa$. However, we currently do not take a very careful analysis of it. We have achieved a $\kappa^{4/3}$ dependency. We think it is possible to improve this dependence using the technique in Cao et al. (2019) (On explicit $ L^ 2$-convergence rate estimate for underdamped Langevin dynamics). And we will work on it in the near future.
>
> ● In Lemma 5.1, it is unclear to me what it means that "$x_n$ and $x_n^∗$ are coupled synchronously". Do you mean that they are driven by the same Brownian process?
> It exactly means that they are driven by the same Brownian motion.
>
> ● Page 9, Lines 277-281: I am confused by this paragraph. By definition, isn't the random weight $w_n(s,\alpha)$ a $d$-dimension vector? If so, how can you apply Claim (A) in Lemma 5.2?
>
> Thanks for pointing out the ambiguity. Line 277 illustrates how our choice of $\rho$ and $\rho’$ inherits the midpoint discretization. And Claim (A) shows that the match of expectation is critical for obtaining this acceleration. In rigorous proof, we apply (A) of Lemma 5.2 with $F(s) = \langle w_n(0, \alpha), \nabla f(x_n(s)) - \nabla f(\hat{x_n}(\alpha)) \rangle$. And this analysis neglects a discretization error on $\bar{w}$ and $w_n(s, alpha)$ as discussed in line 280. And it is supposed to be an inner product in line 279. We will make it clear in a further revised version.
>
>
>
> ● Page 12, (A.1): I am not sure why the authors introduce the extra parameter $\gamma$. As far as I can see, $\gamma$ is fixed as $2$ in the rest of the proof.
>
> Thanks for raising this issue with us. We indeed fix $\gamma = 2$ in the rest of the proof. We will specify that $gamma = 2$ in a further revised version.
>
> ● Page 14, Section B.1.1: I don't see why the random process $B_t^{\alpha}$ is a Brownian bridge. Is it supposed to be the random process $B_t$ conditioned on $B_{\alpha}$? And why do we need to introduce this process in the first place?
>
> $B_t^s$ is a Browning ridge which is conditioned on $B_{\alpha}$. And the process is the SDE of the Brownian bridge. We introduce $B_t^s$ for technical considerations. The diffusion process in line 428 is an adapted process conditioned on $B_{\alpha}$. And process $(x_n(t), v_n(t))$ driven by $B_t^{\alpha}$ is not adapted to the filtration when $t < \alpha$.
>
>
> ● Page 14, the equations under Line 431: Here the authors exchange the order of differentiation and expectation, which is not justified. Indeed, it is not even clear if $\Omega(t)$ is differentiable since it involves the solution trajectories of SDEs.
>
> Thanks for raising this to us. The exchange of differential and expectation requires further validation. And a rigorous proof will be considering the evolution of the process not in expectation but only $\mathrm{d} \Omega(t)/\mathrm{d} t$ and then taking the expectation in line 437. And in line 437, we take the differential of $\Omega(t)$ using Ito’s formula.
>
>
> ● The convergence rates reported in the introduction are inconsistent with the ones in Table 1. Specifically, the rate by [Shen and Lee, 2019] should be $\tilde{O}\left( \frac{d^{1/3}}{\epsilon^{2/3}} \right)$ (Page 2, Line 47).
>
> Thank you for pointing out this issue. It is indeed $\epsilon^{2/3}$ in the denominator.
>
> ● And the rate by [Freund et al., 2022] should scale linearly with $O\left( \frac{1}{\epsilon} \right)$ (Table 1, Row 5).
>
> We compare the convergence rate in Wasserstein distance whereas the $\frac{1}{\epsilon}$ rate is proven in KL distance, which implies a $\frac{1}{\epsilon^2}$ rate in Wasserstein distance.

---

> > ### Comment · Reviewer_TP8b · 2023-08-16
> >
> > I thank the authors for their response. It addresses some of my questions, but some parts remain unclear and ambiguous to me. In particular, I still don't have a good understanding why the random step size enables one to bound the discretization error $\\|\int_0^t x_n(s) - x_n^*(s) ds\\|^2$ in terms of $\mathbb{E} \Omega_n(t)$, which seems to be the key to obtaining dimension-free bounds.
> >
> > After reading other reviews, I remain overall positive about the paper and decide to keep my score. On the other hand, the presentation of the paper has much room for improvement, and I strongly encourage the authors to include more detailed explanations on the key techniques and highlight the necessity of randomized stepsize scheme in the revision.

---

### Decision · Program_Chairs · 2023-09-21

**Decision:**

Accept (poster)

**Comment:**

The paper proposes a Langevin Monte Carlo variant to sample from a strongly log-concave target distribution with a composite structure. Authors are inspired by Randomized Midpoint Method and establish sample complexity in Wasserstein-2 distance. Authors achieve this by only discretizing the smooth convex part of the negative log likelihood. Resulting algorithm enjoys faster rates.

The paper is reviewed by 4 reviewers with the following Rating/Confidence scores: 5/4, 7/3, 6/4, 6/2. The methodology is an interesting extension of Randomized Midpoint method and improvements are significant. I recommend including this paper to conference proceedings.